# Cultural stigma, psychological distress and help-seeking: Moderating role of self-esteem and self-stigma in Lebanon

Myriam El Khoury-Malhame[1]*, Toni Sawma[1], Souheil Hallit[2,3,4], Chloe Joy Younis[1], Jad Jaber[1], Rita Doumit[5]

1 Department of Psychology and Education, School of Arts and Sciences, Lebanese American University, Beirut, Lebanon, 2 School of Medicine and Medical Sciences, Holy Spirit University of Kaslik, Jounieh, Lebanon, 3 Psychology Department, College of Humanities, Effat University, Jeddah, Saudi Arabia, 4 Applied Science Research Center, Applied Science Private University, Amman, Jordan, 5 Alice Ramez Chagoury School of Nursing, Lebanese American University, Byblos, Lebanon

* myriam.malhame@lau.edu.lb

## Abstract

### Background/Objective

Mental illness is a common and often stigmatized condition. Stigma around mental illness refers to the negative attitudes and beliefs society holds about individuals with mental health conditions and can sometimes prevent those individuals from seeking adequate therapy. This study aims to explore the intricate relation between stigma, psychological distress, self-esteem, and help-seeking attitudes among young adults with diagnosed mental illnesses and further investigates the moderating effect of self-esteem and self-stigma.

### Method

A cross-sectional online survey was shared via digital platforms between February 2023 and August 2024. A final sample of 245 participants with clinical mental health diagnoses, from predominantly bachelor-level backgrounds and located in different Lebanese regions, participated in the study and filled demographics data as well as assessments for stigma (Stigma Scale), self-esteem (Rosenberg Self-Esteem scale), help-seeking(Attitudes Towards Seeking Professional Psychological Help-short form (ATSPPH-SF)), distress (Kessler Psychological Distress scale (K6)), and self-stigma (Self-Stigma Questionnaire (SSQ)).

### Results

Results revealed that self-esteem and self-stigma moderated the association between stigma, psychological distress, and help-seeking attitude. At low and moderate levels of self-esteem, higher psychological distress was significantly associated

**Data availability statement:** Data are currently attached as supplement information in an excel sheet.

**Funding:** The author(s) received no specific funding for this work.

**Competing interests:** The authors have declared that no competing interests exist.

with lower help-seeking attitude, while at moderate and high levels of self-stigma, higher psychological distress was associated with lower help-seeking attitude.

## Conclusion

Stigma remains a pervasive condition closely related to increased psychological suffering, decreased self-esteem and lower tendencies to seek help. These inaccurate beliefs and stereotypes are associated with overall discomfort in people with mental disorders perceived as unpredictable, impulsive, and dangerous. It is important for public policy makers within collectivist cultures to better address these misconceptions and promote help-seeking attitudes.

## Background

In addition to the challenges of suffering from mental disorder(s), individuals often must cope with the stigma that emerges from it as mental health stigma remains a widespread social problem impacting people's well-being. Stigma is associated with shame, embarrassment, and rejection that in turn are linked to individuals being avoided or dismissed from others [1,2]. When a phenomenon is stigmatized, a particular individual or group is labeled based on misconceptions and false stereotypes [3]. This in turn alters public beliefs and impressions regarding a specific person or community [4]. Furthermore, individuals who are subjected to stigma can experience loss of purpose and ambitions, loss of friendships and relationships as well as diminished quality of life and work opportunity threats [5]. The stigma associated with a particular condition, such as mental illness, can be associated with social exclusion and discrimination [6].

### Stigma, self-stigma and mental distress

A significant body of research on mental health stigma and its deleterious impacts has explained how the perception of individuals with schizophrenia as dangerous and commonly crazy has led to internalized negative beliefs about oneself referred to as self-stigma [7]. In Lebanon, a Middle Eastern country, university students with mental illnesses further express emotions of shame and embarrassment, leading to their marginalization and social isolation [8]. Karam et al. (2019) highlight that people who suffer from mental disorders in Lebanon face major challenges when it comes to accessing mental health services and seeking psychological treatment, because of the stigmatized association of seeking psychological help with weakness, inadequacy and inability to deal with one's own challenges [9]. As such, when individuals internalize societal perceptions, isolate themselves and feel embarrassed and ashamed of their mental illness, they are self-stigmatizing themselves and further reinforcing the negative loop cultural stigma generates. There is a clear distinction between external stigma (also referred to as public or cultural stigma) and self-stigma (also referred to as internalized stigma), as they are conceptually distinct yet interconnected. Stigma refers to the set of negative stereotypes, prejudices, and discriminatory behaviors

that society holds toward individuals with mental illness [3,6]. It encompasses external, culturally embedded attitudes and practices that devalue individuals based on their mental health status. Self-stigma on the other hand, is a powerful psychological process that refers to the internalization of these negative societal beliefs by individuals themselves. It is often accompanied with diminished self-worth, shame, and feelings of failure [7], and can significantly alter behavior and self-concept [10,11]. Stigma assessments capture the experience of perceived discrimination, reluctance to disclose ill-ness status whereas self-stigma highlights how individuals endorse and absorb negative stereotypes. Although both forms of stigma operate at different levels, social and individual respectively, they are nonetheless known to negatively impact psychological outcomes magnifying the emotional burden of distress and reducing motivation to seek support [12].

## Impact of stigma and self-stigma on self-esteem

Among the most unfortunate outcomes of stigma on mental health is the threat of losing one's self-esteem due to the ongoing instilled belief of being a disappointment and achieving nothing to be proud of [13]. According to numerous stud-ies [14–16], it has been documented that stigmatization is generally associated with lower self-esteem and a diminished sense of worth. This is particularly relevant for people struggling with mental illness; as the stigma around their disorders was systematically reported to correlate with lower self-esteem scores [17]. In other words, individuals who have been diagnosed with psychopathology are frequently labeled as "mentally ill". Their constant exposure to this social stigma might often reinforce their belief in being part of a lower social class, not worthy of love and acceptance and incapable of conforming with others, associated with a subsequent internalization of stigma. The anticipation of being discriminated against correlates with underlying feelings of shame and guilt. This in turn could be associated with self-devaluation, dis-torted self-perceptions, lower self-esteem, and even denial of their sufferings [18].

## Impact of stigma and self-stigma on help-seeking attitudes

Accumulating evidence supports the negative impact of both stigma and self-stigma as a barrier to seeking mental health services in young adults already suffering from mental illness [19,20]. As such, heightened levels of cultural stigmatization are known to associate with poorer outcomes especially for those with pre-existing distress, anxiety, and depression [21]. Patients' decreased self-esteem subsequent to mental health stigma has been recently investigated in relation to rising rates of mental distress, on one hand, and poor health seeking attitude on the other [22,23]. Global reports highlight the discrepancy between exponentially increasing mental diagnoses versus poor treatment-seeking attitudes. Averages indi-cate that although 1 in 4 individuals in Europe have a mental disorder, less than 20% of them go to therapy [24]. Similar data was recently reported in Lebanon after the massive explosion of the Beirut port, whereby more than 80% reported high distress while only 10% of the people sought mental support [25]. This was particularly counterintuitive amidst ongoing outreach campaigns and in spite of psychotherapy being made freely accessible by professionals after the urban disaster. Pervasive social stigma was reported as a major indicator of unwillingness to access or continue mental health care services, in addition to financial considerations [26,27]. Those with mental diagnoses are left navigating challenges like relational complications, professional impairments and personal stress alone [11], thus increasing likelihood of resort-ing to harmful coping mechanisms [28]. Lack of social assistance is observed alongside sadness, anxiety, depression and loneliness [29], damaging altogether physical and mental conditions and prospective prognosis [30]. Social exclusion is also known to decrease tendencies to seek help for stigmatized individuals with mental disorders [31]. Thus, on top of cultural stigma, they must deal with pervasive self-stigma and extra layers of social ostracization.

Minorities and youth are disproportionately deterred by stigma [24]. It could be that for this vulnerable group, stigma extends beyond societal opinions and correlates with one's judgment of one's own self at a critical time of identity forma-tion and significant value for interpersonal relationships [32]. This key relation between perceived stigma and internal-ized personal stigma at this developmental stage might make individuals with mental illnesses adopt more rigid negative attitudes toward themselves [33]. The negative self-perceptions and lower self-esteem associated with stigma inflate

internalized demeaning stigma and further burden young adults and adolescents, significantly raising personal suffering and isolation [34,35]. This creates a negative self-reinforcing loop, posing a barrier to getting professional psychological help or persevering with a chronic treatment [24].

## Focus on the Lebanese context

In as much as mental health stigmatization is reported to be globally widespread, it could be even more pronounced in the global South generally, and in Arab countries particularly. Mental health literacy remains massively neglected, stigmatization remains understudied, and help-seeking hindered by additional cultural factors [36]. Lebanon is a small Levantine country with a unique collectivist culture and a heterogeneous landscape of 18 officially recognized religious groups [37]. Family remains the cornerstone of Lebanese society, characterized by frequent gatherings and a strong sense of collective responsibility. Families gravitate around religious and regional identities, altogether patching a complex sense of social belonging. This provides a strong collective sense of resilience yet heightens the weight of collective expectations, notably regarding mental health and mental health stigma. In this part of the world, external stigma is viewed as a weakness, and accounted for by certain religious (mis)interpretations, for instance attributing psychological issues to supernatural causes such as possession by an evil spirit or divine judgment or punishment [38]. Seeking support from religious clergy instead of mental health professionals delays diagnosis and treatment [39]. As such, stigma not only hinders early intervention and timely treatment in this context but might also perpetuate self-stigma in a culture of silence amidst a climate of fear of social judgment, labeling, social exclusion, family embarrassment and discrimination [40]. This makes it challenging for individuals to openly discuss their mental health struggles; as they would fear being referred to with derogatory terms like "crazy", "nuts", or "unstable" [41], and they would further fear their parents and families being deemed "unworthy" or "losers" [42]. Therefore, to avoid shame and embarrassment, individuals might tend to manage issues privately or within confounded family spheres to keep matters secret [9].

The reluctance to seek treatment is even more pronounced among Lebanese young men, who are traditionally expected to be strong, with displays of emotional distress frowned upon by the community [43]. In this regard, mental health problems are believed to resolve on their own as men push forward; they are taught they can handle their problems alone, or else that their problems aren't severe enough to seek professional help [9]. The National Mental Health Strategy for Lebanon (2024–2030) indeed highlights stigma and lack of knowledge/awareness as key barriers to mental health care, especially for men [44]. The strategy documents that poverty, with recent skyrocketing levels, has also taken a significant toll in the country, significantly reducing access to healthcare. Mental health services in Lebanon are thus scarce and fragmented, and at times, fail to meet rising treatment demands [45]. Although mental health services are a pressing matter for the Lebanese government, the allocated budget does not exceed 5% of the general health budget; the government relies on international support through local NGOs to be able to provide no-cost psychosocial and psychological support programs, along with awareness-raising sessions at the community level [46]. Additionally, while approximately only half of the population in Lebanon is insured by private companies [47], most of these insurance plans do not cover mental health services. This leaves Lebanese citizens seeking psychological and psychiatric services needing to expense treatment costs out of pocket.

In Lebanon, treatment but also research on the influence of stigma and self-stigma on mental health and help-seeking behavior is scarce, although the country is going through major collective protracted crises. It is noteworthy that the protracted adversities have taken a toll on Lebanese youth mental health [25], amidst unprecedented sociopolitical instability [48]. Lebanese with mental health issues reportedly feel stigmatized because of the cultural misconceptions around mental health [49]. Stigma around mental health is associated with higher levels of distress among these young adults as they document significant levels of prejudice and greater feelings of despair and anxiety [50]. This constant social marginalization is associated with internalized self-stigma coupled with negative attitudes toward seeking assistance [51], and reported detrimental effect on their self-esteem [40].

Taken together, all the above reviews suggest an intricate relationship between the overall prevalent external stigma associated with mental health and the internalization of stigma, or self-stigma, especially for individuals with mental health disorders. This complex interaction could be best understood within the framework of the stigma management theory that postulates that individuals manage pervasive stigma using various underlying strategies such as concealing and minimizing the stigmatized condition, downplaying its impact or even resisting it [52]. Since self-esteem is a crucial psychological resource that shapes how individuals perceive themselves and respond to social adversity, including stigma, it could be that higher self-esteem would be linked to greater self-efficacy. Conversely, individuals with low self-esteem may be more vulnerable to the detrimental effects of stigma, leading to greater psychological distress and reduced openness to seek professional help. Considering this same theoretical approach in the Lebanese context, self-stigma would not merely reflect passive acceptance of social judgment; it would actively moderate how distress is experienced and whether individuals engage with or avoid professional care. This effect is particularly critical in collectivist cultures, where mental illness can be seen as a threat to family reputation, making internalized stigma even more psychologically disruptive [39,40]. This study thus aims to explore the intricate relation between stigma, psychological distress, self-esteem and help-seeking behavior among young adults with diagnosed mental illnesses. To enhance the normative value and clarity of the research hypotheses, we present them as follows:

H1: Self-esteem will moderate the relationship between psychological distress and help-seeking attitudes, such that the negative association between psychological distress and help-seeking will be weaker at higher levels of self-esteem.

H2: Self-stigma will moderate the relationship between psychological distress and help-seeking attitudes, such that individuals with higher self-stigma may show less favorable help-seeking attitudes when experiencing psychological distress.

## Methods

### Ethical approval

In accordance with the Lebanese American University's research protocol, the Institutional Review Board IRB reviewed and approved this study (Tracking number LAU.SAS.MM2.8/Mar/2023). Participants electronically signed informed consents before enrolling and filling the questionnaire.

### Procedure

The cross-sectional survey was conducted between February 1, 2023 and August 31, 2024. The survey was made constantly available between these time points. Participants received the survey via a Google Form shared as a clickable link initially disseminated via personnel social media platforms of primary researchers and their trained assistants (including WhatsApp, Instagram and LinkedIn which are preferred common means of communication in Lebanon). No institutional platform was involved. The link was subsequently sponsored by targeting Lebanese users aged 18–40. Snowballing technique was encouraged and convenience sampling was used by explicitly asking participants at the end of the survey to share the survey link with their own online networks to facilitate organic recruitment. Participants were first asked to approve the informed consent by clicking an approve button after reading the ethical consideration detailing the study's purpose. Participants were given additional contact resources in case of distress and were introduced to anonymity and confidentiality measures. Data integrity was ensured by restricting one response per user and identifying information, such as names were not collected. Participants then completed a series of questionnaires for 15–20 minutes. Only participants who were 18–40 years, living in Lebanon and diagnosed with at least one mental illness were eligible to participate. Eligibility was manually checked by researchers.

## Participants

A total of 259 participants enrolled in the study. The number of participant responses that were excluded from the analysis was 14 due to largely incomplete survey responses. The final sample consisted of 245 participants, with 147 women (60%) and from predominantly bachelor-level backgrounds. Participants were located in different Lebanese governorates and also professionally diagnosed with various types of mental illnesses.

## Measures

In addition to demographic data, participants were asked to fill the following scales:

*The Stigma Scale (KSS)* King et al. (2007) developed the 28-item stigma scale to measure the stigma of mental health. It is organized into three subscales: discrimination (12 items), disclosure (11 items), and potential positive elements of mental illness (5 items). It is a five-point Likert scale ranging from 1(strongly agree) to 5 (strongly disagree), with higher scores indicating greater stigma and with subfactors highlighting the major struggles of patients [53]. The discrimination subscale measures the level of prejudice, exclusion, and discrimination experienced by individuals who suffer from mental illnesses. Some items include examples of discriminatory conduct such as "I worry about telling people I receive psychological treatment". The disclosure subscale measures the degree to which people with mental illnesses reveal their psychological struggles with statements like "People's reactions to my mental health problems make me keep myself to myself". The final subdivision of the scale measures the extent to which people with mental illnesses are seen as possessing positive traits like creativity and insight. This potential positive element subscale includes questions such as "My mental health problems have made me more accepting of other people". The scale shows strong internal consistency and test-retest reliability with a Cronbach's alpha of 0.87 for the discrimination subscale, 0.85 and 0.64 for the disclosure and potential positive elements subscales respectively.

*Self-Stigma Questionnaire SSQ*: The 14-item questionnaire was developed by Ochoa et al. (2015) to evaluate the degree to which people with mental illnesses internalize stereotypes and misconceptions, leading to self-stigmatization [54]. It is built on a shorter version of the original 29-item scale [55]. Each item is evaluated on a 7-point Likert scale, with 1 (strongest agreement) and 7 (strongest disagreement). The scale measures how much individuals have absorbed and instilled unfavorable attitudes and beliefs about themselves and their psychological suffering, such as feelings of inferiority and shame because they have mental disorders. Three aspects of self-stigma are evaluated by the SSQ: internalized stigma, perceived discrimination and stereotypes as well as social functioning, with higher scores indicating lower levels of self-stigma. It has been confirmed to be reliable and effective across a range of populations and languages. The scale systematically shows high level of internal consistency with Cronbach's alpha coefficients ranging from 0.75 to 0.901.

*The Rosenberg Self-Esteem Scale* [56]. This scale consists of 10 items measuring self-esteem. It is a 4-point Likert scale from 0 (strongly agree) to 3 (strongly disagree). Five items are positively worded and the other five are negatively worded. The scale ranges from 0–30. Scores between 15 and 25 are within normal range; scores below 15 suggest low self-esteem. So higher scores suggest higher levels of self-esteem and self-worth. The scale has a Cronbach's alpha of 0.84.

*Attitudes Towards Seeking Professional Psychological Help Scale-short form (ATSPPH-SF)* [57]. It assesses social perceptions towards seeking psychological assistance. It includes 10 items on a 4-point Likert scale with statements such as "Personal and emotional troubles, like many things, tend to work out by themselves". Responses range from 0 (disagree) to 3 (agree). Higher score indicates a more optimistic attitude and higher tendencies to ask for psychological aid. The scale had a Cronbach's alpha of 0.84 among an Arab-speaking population [58].

*Kessler Psychological Distress Scale* [59]: A popular 6-item questionnaire used to measure psychological distress. It evaluates how frequently individuals have felt anxious, hopeless, or unworthy over the past 30 days. Each item is graded using a 5-point Likert scale, with the lowest score being 0 (none of the time) and 4 being the highest (all of the time). The mean total score was computed, with its values varying between 0 and 4. The K6's internal consistency and reliability have been found to be high, with alpha coefficient being 0.86 [60].

## Statistical analysis

The SPSS software version 25 was used for the statistical analysis. The help-seeking attitude score was considered normally distributed since the skewness (=.001) and kurtosis (= −.702) values were between −1 and +1 [61]. A Student's t-test was used to compare two means, the ANOVA test to compare three or more means and the Pearson test to correlate two continuous variables. The moderation analysis was conducted using PROCESS MACRO (an SPSS add-on), version 3.4, Model 1. Self-esteem, stigma and self-stigma scores were included as moderators between psychological distress and attitude help seeking. Interaction terms were probed by examining the association of the predictor with the dependent variable at the mean, one standard deviation below the mean, and one standard deviation above the mean of the moderator. Results were adjusted for all variables that showed a $p < .25$ in the bivariate analysis. A p-value of less than.05 was deemed statistically significant.

## Results

### Sociodemographic data and other characteristics of the sample

Two hundred forty-five young adults with diagnosed mental disorders participated in this study. Most of the participants were aged between 20–29 years (89.8%) and had bachelor's degree (74.7%).

Socio-demographic information as well as mean/standard deviations of used questionnaires are reported in Table 1.

In our sample, more than 60.0% reported having clinical anxiety and stress-related disorders, 22.0% had mood and personality disorders, 10.6% had OCD, and 6.5% were diagnosed with neurodevelopmental disorders, 5.7% with sleep disorders, and 5.7% with other psychiatric problems. Some participants reported comorbid disorders, i.e., they were diagnosed with more than one mental disorder.

**Table 1. Sociodemographic data and other characteristics of the sample (N = 245).**

| Variable | N (%) |
|---|---|
| Age (years) | |
| 18–19 | 16 (6.5%) |
| 20–29 | 220 (89.8%) |
| 30–39 | 7 (2.9%) |
| 40 | 2 (.8%) |
| Sex | |
| Male | 98 (40.0%) |
| Female | 147 (60.0%) |
| Education | |
| High school | 14 (5.7%) |
| Bachelor | 183 (74.7%) |
| Master | 46 (18.8%) |
| PhD | 2 (.8%) |
| | **Mean ± SD** |
| Stigma (mean score) | 3.54 ± .81 |
| Self-esteem | 24.78 ± 2.37 |
| Help-seekingAttitude (mean score) | 2.26 ± .74 |
| Psychological distress (mean score) | 3.06 ± .93 |
| Self-stigma (mean score) | 2.93 ± 1.66 |

## Bivariate analysis of factors associated with help-seeking attitude

The results showed that older participants and those with higher education compared to the other categories had significantly higher help-seeking attitude (Table 2). The Bonferroni post-hoc analysis results showed that, in terms of age, there was a significant difference between the group aged 20–29 years and the following age groups: 18–19 years (p = 0.002), 30–39 years (p = 0.032).

In terms of education, there was a significant difference between high school participants and those with a bachelor degree (p < 0.001) and those with a master degree (p = 0.003).

Correlation analyses revealed higher stigma and psychological distress were significantly associated with lower help-seeking attitude, whereas having higher self-esteem and self-stigma were significantly associated with higher help-seeking attitude (Table 3).

## Moderation analyses

Two linear regressions were conducted. The first examined the help-seeking attitude (dependent variable), psychological distress (independent variable), and self-esteem (moderator). The second examined the help-seeking attitude (dependent variable), psychological distress (independent variable), and self-stigma (moderator). The moderation models were adjusted for the following covariates: age, gender and education. The interaction psychological distress

**Table 2. Bivariate analysis of factors associated with help seeking attitude.**

| Variable | Mean ± SD | t/ F | df/ df1, df2 | P | Effect size |
|---|---|---|---|---|---|
| Age (years) | | 8.88 | 3, 241 | **<.001** | .100 |
| 18-19 | 2.84 ± .43 | | | | |
| 20-29 | 2.18 ± .73 | | | | |
| 30-39 | 2.94 ± .57 | | | | |
| 40 | 3.55 ± .64 | | | | |
| Sex | | −1.43 | 243 | .155 | .186 |
| Male | 2.17 ± .67 | | | | |
| Female | 2.31 ± .78 | | | | |
| Education | | 10.82 | 3, 241 | **<.001** | .119 |
| High school | 3.12 ± .64 | | | | |
| Bachelor | 2.15 ± .77 | | | | |
| Master | 2.37 ± .30 | | | | |
| PhD | 3.40 ± .42 | | | | |

Numbers in bold indicate significant p values. Effect size refers to Cohen's d for sex (d values of .02 = small, .05 = medium, .08 = large) and eta-squared η2 (η2 values of .01 = small, .06 = medium, .14 = large) for age and education.

**Table 3. Correlations of continuous variables with help-seeking attitude.**

| | 1 | 2 | 3 | 4 | 5 |
|---|---|---|---|---|---|
| 1. Attitude help seeking | 1 | | | | |
| 2. Stigma | −.60*** | 1 | | | |
| 3. Self-esteem | .48*** | −.45*** | 1 | | |
| 4. Psychological distress | −.41*** | .75*** | −.31*** | 1 | |
| 5. Self-stigma | .48*** | −.85*** | .48*** | −.72*** | 1 |

*p < .05; **p < .01; ***p < .001

by self-esteem was significantly associated with help-seeking attitude (Beta = .08; t = 4.07; p < .001; 95% CI.04;.12) (Table 4, Model 1 and Fig 1). This suggests that individuals with varying levels of self-esteem may differ in their likelihood to seek help when experiencing psychological distress. At low (Beta = −.44; p < .001) and moderate (Beta = −.25; p < .001) levels of self-esteem, higher psychological distress was significantly associated with lower help-seeking attitude (Table 5, Model 1).

Also, the interaction psychological distress by self-stigma was significantly associated with help-seeking attitude (Beta = .07; t = 2.68; p = .008; 95% CI.02;.12) (Table 4, Model 2 and Fig 2). This suggests that individuals with varying levels of self-stigma (reflected by SSQ scores, where lower scores indicate higher self-stigma) may differ in their likelihood to seek help when experiencing psychological distress. At low (high self-stigma) (Beta = −.28; p = .001) and moderate (moderate self-stigma) (Beta = −.17; p = .011) levels of self-stigma, higher psychological distress was significantly associated with lower help-seeking attitude (Table 5, Model 2).

**Table 4. Moderation analyses: Help-seeking attitude as the dependent variable.**

|  | Beta | T | P | 95% CI |
|---|---|---|---|---|
| Model 1: Self-esteem as the moderator (Standard R² = 0.437; F = 18.15) | | | | |
| Psychological distress | −2.19 | −4.49 | **<.001** | −3.15; −1.23 |
| Self-esteem | −.11 | −1.94 | .054 | −.23;.002 |
| Interaction psychological distress by self-esteem | .08 | 4.07 | **<.001** | .04;.12 |
| Model 2: Self-stigma as the moderator (Standard R² = 0.360; F = 13.14) | | | | |
| Psychological distress | −.37 | −3.34 | **.001** | −.58; −.15 |
| Self-stigma | −.03 | −.44 | .658 | −.18;.11 |
| Interaction psychological distress by self-stigma | .07 | 2.68 | **.008** | .02;.12 |

Numbers in bold indicate significant *p* values.

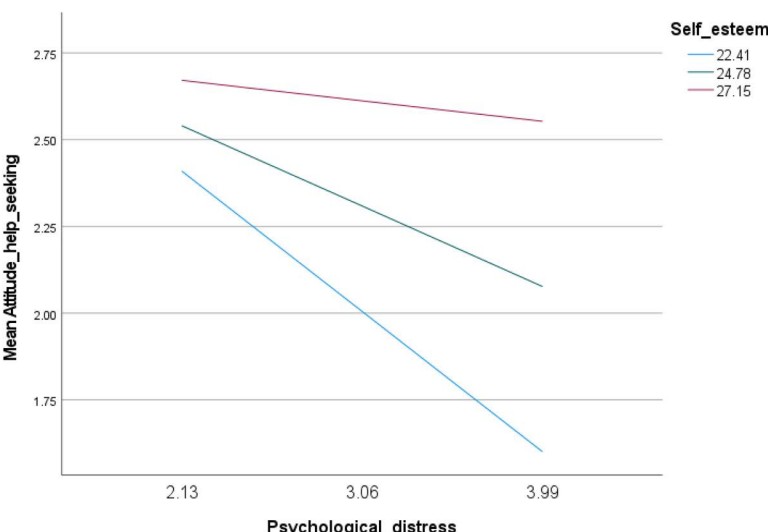

**Fig 1. Plot of the moderating effect of self-esteem on the relation between psychological distress and help- seeking attitude.**

**Table 5. Conditional effects of the focal predictor (psychological distress) at values of the moderator.**

| | Beta | T | P | 95% CI |
|---|---|---|---|---|
| Model 1: Self-esteem as the moderator | | | | |
| Low (= 22.41) | −.44 | −6.19 | **<.001** | −.57; −.30 |
| Moderate (= 24.78) | −.25 | −5.52 | **<.001** | −.34; −.16 |
| High (= 27.15) | −.06 | −1.10 | .273 | −.18;.05 |
| Model 2: Self-stigma as the moderator | | | | |
| Low (= 1.27) | −.28 | −3.28 | **.001** | −.45; −.11 |
| Moderate (= 2.93) | −.17 | −2.55 | **.011** | −.29; −.04 |
| High (= 4.59) | −.05 | −.75 | .455 | −.19;.08 |

Numbers in bold indicate significant *p* values. The moderator was divided into three categories (low, moderate and high) at the mean, 1 Standard Deviation (SD) below the mean and 1 SD above the mean.

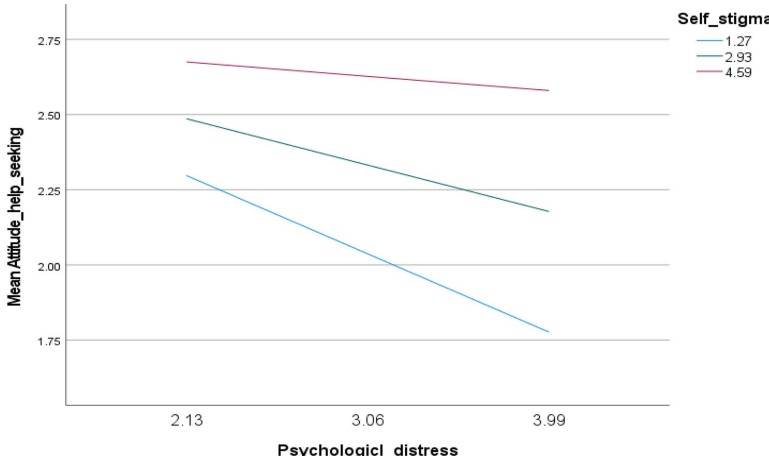

**Fig 2. Plot of the moderating effect of self-stigma on the relation between psychological distress and help-seeking attitude.**

## Discussion

In our study, two forms of stigma, external social stigma as well as self-stigma around mental illness, were studied in relationship to psychological distress, self-esteem and help-seeking attitudes in an underexplored Levantine context. Results from young adults diagnosed with mental health disorders in the Lebanese community show that self-esteem and self-stigma moderated the association between psychological distress and help seeking attitudes. As such, stigmatization, both social and internalized, is systematically associated with decreased help-seeking. The relationship between stigma, psychological distress and help-seeking is further influenced by self-esteem and self-stigma. These findings are best understood in line with the stigma management theory positing that individuals facing external stigma engage in various coping strategies, ranging from avoidance to advocacy, to dampen its negative impacts. In this framework, individuals with mental distress who have high self-esteem would use their personal character strength to override societal stigma and seek help anyway, even if frowned upon by the collectivity. Our data shows they are also more likely to seek professional support if they manage not to internalize pervasive stigmatization.

## Moderating effect of self-esteem between psychological distress and help-seeking attitude

Our results revealed that self-esteem moderated the association between psychological distress and help-seeking attitude at low and moderate levels of self-esteem. This indicates that individuals with low to moderate self-esteem, who are already vulnerable to psychological distress, are less likely to seek help for their mental health concerns. In our study, the negative association between increased psychological distress and lower help-seeking attitude highlights the detrimental role of mental health difficulties in delaying access to treatment. This finding is even more exacerbated for individuals with low self-esteem who might feel more shame, guilt, and inadequacy, leading them to internalize their struggles, rather than reaching out for help. In fact, individuals with low self-esteem are less likely to seek professional help, even when mental health services are accessible and affordable [10]. They might believe they are "weak" or "defective" for experiencing distress, and seeking help could feel like an admission of failure, further eroding their already fragile self-worth. Individuals with low self-esteem may also be more vulnerable to the detrimental effects of stigma, leading to greater psychological distress and reduced openness to seek professional help. Mental health stigma is indeed documented to solidify barriers to seeking mental health recovery in occidental [62] as well as oriental societies [63] as people suffering from mental illnesses often fear prejudice and discrimination when seeking treatment, compounded by a perceived inability to stand up to social pressure. At its worst, in the absence of professional support, heightened levels of stigmatization can increase self-stigma, decrease self-worth and lead to hopelessness and helplessness for individuals with mental distress [64]. This however does not need to be a fatality as individuals with high levels of self-esteem and self-worth generally describe help-seeking as empowering and beneficial rather than a sign of weakness and vulnerability [65]. University students with higher self-esteem seem to be keen on promptly recognizing and addressing issues related to mental health for better outcomes and faster recovery [66]. Subsequently, our results underscore the importance of promoting self-esteem and addressing stigma to encourage help-seeking behaviors among those young adults with mental disorders.

## Moderating effect of self-stigma between psychological distress and help-seeking attitude

This study also illustrates the role of self-stigma in moderating the relationship between psychological distress and help-seeking attitude. Self-stigma refers to internalization of cultural mental health stigmas. It commonly instills feelings of disgrace and disappointment in young individuals regarding their families. Adopting negative perceptions and stereotypes, makes feelings of worthlessness, shame and guilt emerge, reinforcing unfavorable perceptions toward mental illnesses [12]. Additionally, in our research, the interaction effect between self-stigma and psychological distress underscores the complexity of help-seeking attitudes. In other terms, individuals experiencing high psychological distress are less likely to seek help if they also have moderate to high levels of self-stigma. This finding suggests that self-stigma can amplify the negative impact of psychological distress on help-seeking attitudes. As self-stigma increases, individuals may become more reluctant to acknowledge their mental health problems, fear negative judgment, and avoid seeking necessary support. Within this framework, we hypothesize that low self-stigma may serve as a protective factor, mitigating the negative effects of psychological distress on help-seeking attitudes. Conversely, high self-stigma appears to exacerbate this negative relationship, making individuals less likely to seek help when distressed. In our sample, individuals with lower internalized stigma may be more resilient to the negative impacts of stigma and distress, making them more likely to seek help [67]. These positive attitudes may be further reinforced by supportive interactions with healthcare providers.

In line with the Stigma Management Theory (SMT), these findings suggest that stigma shapes individuals' beliefs. First, it influences their perception of the inevitable consequences of mental health issues. Second, it impacts their belief in their own incapacity to overcome challenges, which correlates with poorer outcome expectancies regarding help seeking. This explanation is in line with recent data pooled in the same context among Lebanese individuals with mental illnesses,

whereby participants reported lower levels of self-esteem and quality of life, as well as higher levels of social isolation when subjected to stigma [68]. This reinforces marginalization and social isolation and perpetuates prevalent stigma and discrimination [69]. In the Lebanese context, self-stigma is still a prevalent challenge and mental health is still frowned upon [70], pushing those with psychiatric disorders to silence their struggles and limit their interactions with others, further prolonging the existing social stigma that remains unchallenged.

Within the same SMT framework, self-esteem and self-stigma act as a buffer against cultural stigmatization. Individuals with higher self-esteem and self-affirmation can maintain a positive self-image despite societal disapproval, which in turn mitigates their psychological distress and increases their likelihood of seeking support. On the contrary, self-stigma would amplify cultural stigma. Individuals who internalize social prejudice would alternatively partake in fear of judgment and avoid challenging cultural beliefs, therefore experiencing heightened shame and self-blame leading to increased psychological distress and decreased help-seeking.

## Limitations and future research

To the best of our knowledge, this is the first study addressing stigma, self-stigma and self-esteem in a sample of young adults with existing mental disorders. The modest sample size limits robustness when it comes to moderation analyses. Although rigorously performed, increasing the sample size could benefit in more reliable statistics and more adequate generalization of the findings. Our study indeed focused on educated young adults and should thus be generalized with care. Future studies should consider involving individuals with varying levels of education and broader age ranges. Qualitative reports specifically targeting the Lebanese cultural context could provide additional insights into sensitive aspects of stigma. This approach would also better encompass the diverse experiences within different Lebanese communities considering factors such as gender, religious affiliations and urban versus rural settings. Conducting a longitudinal design and including therapy outcome could also track potential fluctuations in stigma, self-stigma, self-esteem, psychological distress and help-seeking tendencies across several mental conditions. Lastly, it could be useful to track the psychiatric diagnoses of participants to overcome some social desirability biases when self-reporting mental disorders.

## Implications and conclusions

**Practical implications.**  This study sheds light on the significance of mental health stigma and self-esteem on psychological distress and help-seeking attitudes. Findings highlight the pivotal role of both personal and social targets in addressing the matter, especially given that self-esteem and self-stigma were found to moderate the above intricate relationships. In a collectivist Lebanese context, this means capitalizing on social connectedness by intensifying early awareness within appropriate cultural norms.

At a local level, contact with community leaders, known as Mukhtars, and religious figures to address stigma, is essential to bridge the gap between the Non-Governmental Organizations (NGOs) mental health workers and communities. Those can be seen as outsiders who come to change the traditional norms, especially in a sectarian country such as Lebanon [39]. To this extent, culturally sensitive interventions that promote mental health literacy, and challenge misconceptions should be advocated along local leaders including religious clergy; whom have the trust of the community and have a crucial role in referring those who might initially seek help from religious figures or attribute mental health issues to spiritual factors [39]

Other actionable steps would include actively fostering youngsters' self-esteem within families, societies, schools, universities and municipalities with specific evidence-based programs aimed at improving self-esteem and reducing stigma. One such program significantly enhanced emotional intelligence skills in public high-school students, while also targeting self-esteem and self-stigma [71]. Focus group interviews conducted after the workshop showed that the program

effectively instilled favorable self-attitudes and beliefs about addressing psychological distress. In particular, the program helped participants manage and overcome feelings of inferiority and shame concerning mental disorders and individuals experiencing them.

Another practical measure involves using social media platforms such as WhatsApp and Instagram to facilitate open dialogue about mental health and build public empathy. This approach has been supported by studies in stigmatized collectivists nations such as Lebanon [39] and Malaysia [72]. To this extent, people who might have depression or experience anxiety can connect to other Lebanese people, and to mental health specialists and in turn override feelings of loneliness, isolation, and helplessness.

**Theoretical Implications.** The present study provides important theoretical insights into the interplay between stigma, help-seeking behavior, and psychological distress within a collectivist cultural context. Our findings substantiate and extend Stigma Theory by providing evidence that both public and internalized stigma are not only prevalent but deeply interwoven with self-perceptions and psychological suffering in such contexts. The moderating roles of self-esteem and self-stigma confirm key tenets of Stigma Management Theory, particularly the idea that personal resources and internalized beliefs shape individuals' coping strategies in the face of societal rejection.

Finally, in collectivist cultures like Lebanon, where family honor and communal belonging heavily influence identity, our findings suggest that inasmuch as social values offer community support [39], they can negatively influence help-seeking as stigma carries a heightened threat in some social circles. To avoid worsening such internal conflicts, this study urges culturally adapted theories of mental health behavior that account for communal interdependence and honor-based identity systems. Promising success of interventions building on comedy and group cohesion such as drama therapy have also shown efficacy and cultural sensitivity in coming together to heal and overcome stigma [73]. These would be particularly relevant as they would factor in cultural resources such as the reported collective sense of humor, used to laugh at shared miseries and to cope with the country's latest socio- economic and political tensions.

## Conclusion

This study contributes to the limited but growing literature on mental health stigma in the Arab world by examining how stigma, psychological distress, and individual-level moderators such as self-esteem and self-stigma shape help-seeking attitudes in a Lebanese sample. Findings could be relevant in the Global south faced with protracted crises and in the Global North with incrementally rising rates of mental distress. We build on recommendations from the Lancet Commission on ending stigma and discrimination in mental health (2022), suggesting nuanced internal processes may either buffer or compound the effects of stigma. Since approaching people with direct lived experiences of psychological disorders, our research could inform policy makers in comparable Levantine cultures to increase help-seeking attitudes by capitalizing on positive mental health literary to build self-esteem and contribute to individual resilience, while incentivizing proactive efforts to collectively destigmatize and treat mental illness. This would lead to favorable outcomes for both mental and physical health of vulnerable young adults.

## Supporting information

**S1 Data. SPSS data.**
(XLSX)

**S2 File. Supplementary table S2 Table1**. Linear regression with help-seeking attitude (dependent variable), psychological distress (independent variable), and self-esteem (moderator).**Supplementary table** S2 Table **2** Linear regression with help-seeking attitude (dependent variable), psychological distress (independent variable), and self-stigma (moderator).
(DOCX)

## Author contributions

**Conceptualization:** Myriam El Khoury-Malhame, Toni Sawma.

**Data curation:** Jad Jaber, Chloe Younis.

**Formal analysis:** Souheil Hallit, Rita Doumit.

**Investigation:** Souheil Hallit, Jad Jaber, Chloe Younis.

**Methodology:** Myriam El Khoury-Malhame, Toni Sawma, Souheil Hallit, Jad Jaber, Chloe Younis, Rita Doumit.

**Supervision:** Toni Sawma, Souheil Hallit.

**Validation:** Souheil Hallit.

**Visualization:** Souheil Hallit, Chloe Younis.

**Writing – original draft:** Myriam El Khoury-Malhame, Jad Jaber.

**Writing – review & editing:** Myriam El Khoury-Malhame, Toni Sawma, Souheil Hallit, Chloe Younis, Rita Doumit.

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
