## [Decision Letter · Decision Letter 0]

10 Dec 2024

PONE-D-24-53040Cultural Stigma, Psychological Distress and Help-Seeking:

Moderating role of Self-esteem and Self-stigmaPLOS ONE

Dear Dr. El Khoury Malhame,

Thank you for submitting your manuscript to PLOS ONE. After careful consideration, we feel that it has merit but does not fully meet PLOS ONE’s publication criteria as it currently stands. Therefore, we invite you to submit a revised version of the manuscript that addresses the points raised during the review process.

We look forward to receiving your revised manuscript.

Kind regards,

Lakshminarayana Chekuri, MD, PhD

Academic Editor

PLOS ONE

**Journal Requirements:**

2. Please describe in your methods section how capacity to provide consent was determined for the participants in this study. Please also state whether your ethics committee or IRB approved this consent procedure. If you did not assess capacity to consent please briefly outline why this was not necessary in this case.

3. In the online submission form, you indicated that Data are not currently shared publicly. They will be made available for researchers upon reasonable request.

**Additional Editor Comments:**

Thank you for your scholarly contribution. I'd also like to thank the authors for choosing PLOS ONE to publish your findings from this study. Comments from reviewers and myself are provided below. Please review these comments and I suggest addressing them and resubmit your manuscript. Your timely response would help this study be published and accessible to interested readers across the world. I look forward to reviewing your revised manuscript. I wish you good luck with your future endeavors.

Editor’s comments:

While your title specifically states, “cultural stigma”, the scale that was used to measure stigma in the study participants does not seem to be customized to measure “cultural stigma”. I would recommend removing the word “cultural” from the title.

Reviewers' comments:

Reviewer's Responses to Questions

**Comments to the Author**

1. Is the manuscript technically sound, and do the data support the conclusions?

Reviewer #1: Yes

Reviewer #2: Partly

2. Has the statistical analysis been performed appropriately and rigorously? 

Reviewer #1: Yes

Reviewer #2: Yes

3. Have the authors made all data underlying the findings in their manuscript fully available?

Reviewer #1: No

Reviewer #2: Yes

4. Is the manuscript presented in an intelligible fashion and written in standard English?

Reviewer #1: Yes

Reviewer #2: Yes

5. Review Comments to the Author

**Reviewer #1:**  The manuscript appears to be technically sound, and the data presented generally support the conclusions drawn. The research design and methods are appropriate, and the findings seem logically derived from the data. The authors have provided enough evidence to back their claims, and the study is consistent with the research questions posed. The statistical analysis is generally well-executed, with appropriate methods used to analyze the data. The authors have presented the statistical results clearly, and the significance of key findings is adequately reported. However, one area of concern is that the raw data underlying the findings have not been fully made available. According to the PLOS ONE data policy, it is important that all data be accessible unless there are justified restrictions, such as privacy concerns. It would be helpful if the authors could provide the raw data or clarify any issues regarding data sharing. In terms of presentation, the manuscript is written in clear, standard English, and the overall structure is intelligible. The content is presented in a logical sequence, making it relatively easy to follow for readers, and the language is consistent with academic standards. Overall, while the manuscript is well-written and methodologically sound, addressing the data availability issue would enhance its transparency.

**Reviewer #2: ** The manuscript addresses a highly relevant and globally significant topic. Understanding the mechanisms and variables influencing help-seeking behavior in individuals with mental health disorders is crucial. The analyses conducted and the results obtained reflect findings already established in international literature, but the study provides an interesting perspective on the specific context of Lebanon. While the manuscript addresses a highly relevant topic and contributes to understanding help-seeking behaviors in a specific cultural context, several aspects require substantial revision and clarification to enhance the clarity, rigor, and overall impact of the study. As such, I recommend major revisions before the manuscript can be considered for publication.

I recommend clarifying the geographical or cultural group of interest in the title. This would avoid potential confusion, especially since the manuscript does not include specific measures targeting "cultural stigma," as suggested in the title, making it unclear if the study directly addresses the impact of this construct.

Introduction: the introduction is comprehensive and effectively outlines the variables studied. Given the relevance of the study to the Lebanese context, the authors could emphasize data specific to this national context earlier in the introduction while reducing details on aspects, such as structural stigma, that are not subsequently addressed in the manuscript.

Additionally, the introduction cites a study by El Khoury-Malhame et al. (2024), but the type of trauma referenced is unclear, making its relevance to the manuscript debatable. Finally, most references are relatively dated, with only a few recent sources. I suggest incorporating more up-to-date references.

Methods

The measures section is detailed but omits some important information. For instance, some instruments lack a description of the scoring range, making it difficult to interpret the results. Additionally, two instruments measure stigma, but the distinction between their constructs is unclear (e.g., what specific type of stigma does the The Stigma Scale measure?). Listing these two instruments consecutively could improve the paragraph's readability.

In the statistical analysis section, there are placeholders marked as "(ref.)" where references are missing. Attention should also be given to the formatting of statistical significance (e.g., uppercase or lowercase "p").

Results

The results section raises some concerns. The age distribution is heavily skewed towards the 20–29 age group, with only two participants in the oldest age group. This discrepancy makes it challenging to consider the groups comparable. Revisiting the age classification to create more balanced groups might be helpful.

A similar issue arises with the education levels, where bachelor’s and master’s degree holders constitute the vast majority of the sample. Table 2 reflects these disparities, and the unequal group sizes could undermine the statistical comparisons. It would also be helpful to specify which age or education groups differ significantly from one another rather than only providing the general significance level.

For Table 5, it is unclear how the scores were categorized into "low," "moderate," and "high," and the sample sizes for each group should be provided.

Discussion

The discussion is detailed but occasionally reads like an introduction, as it focuses on external studies rather than interpreting the manuscript’s findings. For instance, the paragraph on "Help-seeking in relation to stigma and psychological distress" includes studies that might be better suited to the introduction. The authors note that older age and higher education levels are protective factors for help-seeking attitudes. However, the data do not fully support this claim. For instance, the 18–19 age group scores higher than the 20–29 group and similarly to the 30–39 group. Revising this paragraph to focus on interpreting the results and exploring possible reasons for the observed age and education distributions would strengthen the discussion.

In the paragraph on "Help-seeking in relation to self-stigma," the authors define stigma as affecting students and their families, but this narrowing of the construct should be clarified—why are the authors only considering students? Does it refer to some characteristics of the sample which are not fully described in the method section? Furthermore, it is unclear why participants with lower self-stigma are less likely to seek help. This counterintuitive result warrants further discussion, but instead, the authors argue that higher self-stigma increases reluctance to seek help, which is not supported by the results. I suggest rewriting this section to focus on interpreting the findings and exploring their implications.

Limitations

The authors could explicitly address the uneven distributions in age and education, which complicate inferences about these variables.

Conclusion

The authors could expand on the potential implications and impact of their findings on the Lebanese context, which would strengthen the significance of the manuscript.

Minor points: There are a few typographical errors throughout the manuscript (e.g page 15: "developped", page 19: "as shown" instead of "has shown") that should be carefully reviewed and corrected. A thorough proofreading is recommended to ensure consistency and accuracy in the text."

6. PLOS authors have the option to publish the peer review history of their article (what does this mean? ). If published, this will include your full peer review and any attached files.

**Do you want your identity to be public for this peer review?** For information about this choice, including consent withdrawal, please see our Privacy Policy .

Reviewer #1: **Yes: ** Hu Jun

Reviewer #2: No

---

## [Author Response · Author response to Decision Letter 1]

25 Feb 2025

Dearest Editor

On behalf of my team, I want to thank you and the reviewers to efficiently invest time to help us uplift our manuscript, all while ensuring scientific rigor.

We are hereby submitting our revisited our manuscript, massively shaped by the recommendations of the team of experts. We acknowledge it has helped us further refine our narratives and hope it meets the publication standards

As such, we are providing below the point-by-point response to each of the 3 reviewers in bold. We have highlighted changes in text when feasible to facilitate tracking.

Looking forward to your feedback

Myriam El Khoury-Malhame

Reviewer #1:

The manuscript appears to be technically sound, and the data presented generally support the conclusions drawn. The research design and methods are appropriate, and the findings seem logically derived from the data. The authors have provided enough evidence to back their claims, and the study is consistent with the research questions posed. The statistical analysis is generally well-executed, with appropriate methods used to analyze the data. The authors have presented the statistical results clearly, and the significance of key findings is adequately reported.

Thank you for the humbling feedback

However, one area of concern is that the raw data underlying the findings have not been fully made available. According to the PLOS ONE data policy, it is important that all data be accessible unless there are justified restrictions, such as privacy concerns. It would be helpful if the authors could provide the raw data or clarify any issues regarding data sharing.

Data cannot be shared publicly due to restrictions imposed by the Ethics Committee, but is available upon request. Data requests may be sent to the corresponding author.

In terms of presentation, the manuscript is written in clear, standard English, and the overall structure is intelligible. The content is presented in a logical sequence, making it relatively easy to follow for readers, and the language is consistent with academic standards.

Much appreciated comment, thank you! 

Reviewer #2:

The manuscript addresses a highly relevant and globally significant topic. Understanding the mechanisms and variables influencing help-seeking behavior in individuals with mental health disorders is crucial. The analyses conducted and the results obtained reflect findings already established in international literature, but the study provides an interesting perspective on the specific context of Lebanon.

This is indeed the objective of the project, to give visibility to the oriental Levantine side of the Global South.

While the manuscript addresses a highly relevant topic and contributes to understanding help-seeking behaviors in a specific cultural context, several aspects require substantial revision and clarification to enhance the clarity, rigor, and overall impact of the study. As such, I recommend major revisions before the manuscript can be considered for publication.

I recommend clarifying the geographical or cultural group of interest in the title. This would avoid potential confusion, especially since the manuscript does not include specific measures targeting "cultural stigma," as suggested in the title, making it unclear if the study directly addresses the impact of this construct.

We have added the geographical group and rephrased the title which now reads: “Cultural Stigma, Psychological Distress and Help-Seeking: Moderating role of Self-esteem and Self-stigma In Lebanon”

Introduction:

The introduction is comprehensive and effectively outlines the variables studied. Given the relevance of the study to the Lebanese context, the authors could emphasize data specific to this national context earlier in the introduction while reducing details on aspects, such as structural stigma, that are not subsequently addressed in the manuscript.

Additional data specific to the Lebanese context was added early on in the introduction whereas it was clustered towards the end of the introduction in the previous version. The detailing of structural stigma was also removed as it is indeed not addressed in the manuscript.

Additionally, the introduction cites a study by El Khoury-Malhame et al. (2024), but the type of trauma referenced is unclear, making its relevance to the manuscript debatable.

We have clarified the choice of reference. The intention was to highlight the finding that stigmatized attitudes towards help seeking are massively preventing people from reaching out to mental support, even in the aftermath of an acute trauma, with ongoing outreach campaigns and availability of funding for mental health initiatives in the country.

Finally, most references are relatively dated, with only a few recent sources. I suggest incorporating more up-to-date references.

Thank you for the mention. The team has completely revisited its references and has updates them systematically.

Methods

The measures section is detailed but omits some important information. For instance, some instruments lack a description of the scoring range, making it difficult to interpret the results.

We have added the omitted information when feasible to make the results easier to interpret. For some scales, there was no defined cutoffs, we nonetheless have highlighted the directionality of scores, for instance with higher scores indicating higher stigma.

Additionally, two instruments measure stigma, but the distinction between their constructs is unclear (e.g., what specific type of stigma does the The Stigma Scale measure?). Listing these two instruments consecutively could improve the paragraph's readability.

Thank you for the mention. We have highlighted the crucial yet understandably subtle differences between:

- Stigma scale: measuring overall stigma around mental health

- Self-Stigma scale: measuring the internalized stigma of one’s own mental illness

Scales were also listed consecutively as suggested.

In the statistical analysis section, there are placeholders marked as "(ref.)" where references are missing.

This has been amended and the empty ref. were removed. Apologies about the mishap.

Attention should also be given to the formatting of statistical significance (e.g., uppercase or lowercase "p").

This has also been amended and fixed throughout the manuscript.

Results

The results section raises some concerns. The age distribution is heavily skewed towards the 20–29 age group, with only two participants in the oldest age group. This discrepancy makes it challenging to consider the groups comparable. Revisiting the age classification to create more balanced groups might be helpful.

The purpose of the study as stated in the introduction was to assess stigma and self-stigma in young adults to avoid inter-generational challenges when it comes to understanding and awareness or acceptability of mental health stigma. In line with this, only individuals 18-40 were included and the 2 “older” participants were removed to avoid skewing the results.

This has been added to the limitation section as well to better investigate potential variability of results with older age groups.

A similar issue arises with the education levels, where bachelor’s and master’s degree holders constitute the vast majority of the sample. Table 2 reflects these disparities, and the unequal group sizes could undermine the statistical comparisons.

As per the above, we have acknowledged this and added the limitations related to both age and education in the limitation section

It would also be helpful to specify which age or education groups differ significantly from one another rather than only providing the general significance level.

Spot on comment. We added the results of the Bonferroni post-hoc analysis as follows:

“The Bonferroni post-hoc analysis results showed that, in terms of age, there was a significance difference between the group aged 20-29 years and the following age groups: 18-19 years (p = 0.002), 30-39 years (p = 0.032) and 40 years and above (p = 0.041). In terms of education, there was a significant difference high school participants and those with a bachelor degree (p < 0.001) and those with a master degree (p = 0.003).”

For Table 5, it is unclear how the scores were categorized into "low," "moderate," and "high," and the sample sizes for each group should be provided.

Thank you for this comment. We added this sentence to the statistical analysis paragraph:

Interaction terms were probed by examining the association of one predictor with the dependent variable at the mean, 1 SD below the mean and 1 SD above the mean of the moderator.

A footnote was also added below Table 5 explaining the above idea.

Discussion

The discussion is detailed but occasionally reads like an introduction, as it focuses on external studies rather than interpreting the manuscript’s findings. For instance, the paragraph on "Help-seeking in relation to stigma and psychological distress" includes studies that might be better suited to the introduction.

Thank you for the suggestion. We have added focus on results analysis and discussion, shifted some references as per the recommendation to the introduction and rephrased or removed redundant explanations. All the remodeling was highlighted for ease of tracking.

The authors note that older age and higher education levels are protective factors for help-seeking attitudes. However, the data do not fully support this claim. For instance, the 18–19 age group scores higher than the 20–29 group and similarly to the 30–39 group. Revising this paragraph to focus on interpreting the results and exploring possible reasons for the observed age and education distributions would strengthen the discussion.

Thank you for the keen observations. We have attempted to discuss this non-linear relationship, highlighting nonetheless the impact of education and age on the overall framework in the Lebanese context. The manuscript now reads “Education and older age could play important yet variable roles in favor of help-seeking behaviors... to be acknowledged and treated”.

In the paragraph on "Help-seeking in relation to self-stigma," the authors define stigma as affecting students and their families, but this narrowing of the construct should be clarified—why are the authors only considering students? Does it refer to some characteristics of the sample which are not fully described in the method section?

Apologies for the confusion. The intent was to mention young adults and not merely students and the proper description has been amended as such.

Furthermore, it is unclear why participants with lower self-stigma are less likely to seek help. This counterintuitive result warrants further discussion, but instead, the authors argue that higher self-stigma increases reluctance to seek help, which is not supported by the results. I suggest rewriting this section to focus on interpreting the findings and exploring their implications.

Thank you for the comment. We have attempted to increase clarity by reformulating the statements and mentioning nonetheless that “Additionally, in our research, the interaction effect between self-stigma and psychological distress underscores the complexity of help-seeking behaviors”

Limitations

The authors could explicitly address the uneven distributions in age and education, which complicate inferences about these variables.

This has been done accordingly. We have added mentions about that “It could also be beneficial to compare findings to those in other age groups: either younger adolescents and/or older adults to better assess the evolution of stigma around mental health as a function of developmental as well as generational, cultural and educational factors. Our study indeed focused on educated young adults and should thus be generalized with care. Future research could also focus on refining the role of education in mitigating or inflating stigma and help-seeking by comparing various levels of education and higher education.”

Conclusion

The authors could expand on the potential implications and impact of their findings on the Lebanese context, which would strengthen the significance of the manuscript.

This has been done accordingly

Minor points: There are a few typographical errors throughout the manuscript (e.g page 15: "developped", page 19: "as shown" instead of "has shown") that should be carefully reviewed and corrected. A thorough proofreading is recommended to ensure consistency and accuracy in the text."

Thorough proofreading has been done, hopefully to the standards of the journal. 

Reviewer #3

Main Issues, Arguments, and Conclusions

This study explores the connections between Cultural Stigma, Psychological Distress, and Help-Seeking Behavior, with a particular focus on the moderating roles of Self-esteem and Self-stigma. Using a cross-sectional online survey of 245 young Lebanese adults, the study found significant negative associations between cultural stigma and both psychological distress and help-seeking attitudes. Moreover, self-esteem and self-stigma affected these relationships, increasing psychological distress and reducing help-seeking intentions when cultural stigma was high and self-esteem was low or self-stigma was high.

The findings highlight that the harmful impact of cultural stigma on mental health is particularly strong in the Lebanese cultural context. The moderating roles of self-esteem and self-stigma provide important insights for designing culturally relevant mental health interventions. The study suggests that mental health interventions should consider cultural differences, focusing on improving self-esteem and managing self-stigma to help individuals cope with stigma-related distress and increase their willingness to seek help.

Positioning and Contributions in the Existing Literature

This study adds value to the existing literature by examining the dynamics of cultural stigma, psychological distress, and help-seeking behavior within the specific cultural context of Lebanon, a non-Western setting that has been underrepresented in mental health stigma research. Unlike much of the existing literature based in Western contexts, this study makes the following key contributions. First, it broadens the cross-cultural perspective on mental health stigma by examining how cultural factors shape stigma and its consequences in Lebanon’s social and cultural environment. The empirical data from this non-Western context enhance cross-cultural comparisons in this field.

Second, the study improves our understanding of how self-esteem and self-stigma influence the effects of cultural stigma on psychological distress and help-seeking behavior. This theoretical insight supports the use of moderation effects when designing targeted mental health interventions.

Finally, the study provides practical suggestions for tackling cultural stigma, particularly for groups with high levels of stigma. By emphasizing the importance of strengthening self-esteem and reducing self-stigma, the study offers useful guidance for policymakers and mental health professionals who aim to design culturally appropriate interventions.

Strengths and Weaknesses

This study has several strengths. It addresses an important issue by exploring how cultural stigma affects mental health and help-seeking behaviors, with a focus on the moderating roles of self-esteem and self-stigma. This focus enhances our understanding of the psychological processes involved in mental health and offers valuable insights for future research and interventions. The study also uses well-established psychological measurement tools, such as the Rosenberg Self-esteem Scale and the Kessler Psychological Distress Scale, to ensure the accuracy and reliability of the data. Furthermore, the study contributes to the broader field of cross-cultural research by exploring cultural stigma in a non-Western context, specifically Lebanon, which fills a gap in the literature.

However, there are areas for improvement. One limitation is the representativeness of the sample. The study primarily focuses on young adults aged 20-29, which may limit the ability to apply the findings to other age groups or social demographics.

---

## [Decision Letter · Decision Letter 1]

21 Apr 2025

PONE-D-24-53040R1Cultural Stigma, Psychological Distress and Help-Seeking:

Moderating role of Self-esteem and Self-stigmaPLOS ONE

Dear Dr. El Khoury Malhame,

Thank you for submitting your manuscript to PLOS ONE. After careful consideration, we feel that it has merit but does not fully meet PLOS ONE’s publication criteria as it currently stands. Therefore, we invite you to submit a revised version of the manuscript that addresses the points raised during the review process.

We look forward to receiving your revised manuscript.

Kind regards,

Lakshminarayana Chekuri, MD, PhD

Academic Editor

PLOS ONE

Journal Requirements:

Additional Editor Comments (if provided):

Thank you for your scholarly contribution. I'd also like to thank the authors for choosing PLOS ONE to publish your findings from this study. Comments from reviewers are provided below. Please review these comments and I suggest address them and resubmit your manuscript. Your timely response would help this study be published and will make it accessible to interested readers across the world. I look forward to reviewing your revised manuscript. I wish you good luck with your future endeavors.

Reviewers' comments:

Reviewer's Responses to Questions

**Comments to the Author**

1. If the authors have adequately addressed your comments raised in a previous round of review and you feel that this manuscript is now acceptable for publication, you may indicate that here to bypass the “Comments to the Author” section, enter your conflict of interest statement in the “Confidential to Editor” section, and submit your "Accept" recommendation.

Reviewer #1: (No Response)

Reviewer #2: All comments have been addressed

2. Is the manuscript technically sound, and do the data support the conclusions?

Reviewer #1: Yes

Reviewer #2: Yes

3. Has the statistical analysis been performed appropriately and rigorously? 

Reviewer #1: Yes

Reviewer #2: Yes

4. Have the authors made all data underlying the findings in their manuscript fully available?

Reviewer #1: Yes

Reviewer #2: Yes

5. Is the manuscript presented in an intelligible fashion and written in standard English?

Reviewer #1: Yes

Reviewer #2: Yes

6. Review Comments to the Author

Reviewer #1: Dear Authors,

Thank you for your careful and thoughtful revisions to the manuscript. It is evident that substantial effort has been invested in improving the paper across multiple sections. Your incorporation of a theoretical framework, expansion of the literature review, clarification of methodology, and inclusion of moderation plots have notably enhanced the overall clarity and contribution of the study.After reviewing the revised manuscript, I find that the majority of the major concerns have been addressed. However, a few important issues remain unresolved or only partially addressed. Below is a summary of these points, organized by manuscript sections and marked by importance level.

1. Introduction / Background Section

1.1 While the narrative logic is coherent, the hypotheses remain embedded within the text. I recommend clearly enumerating them at the end of the Background section (e.g., H1, H2, etc.) for improved clarity and structure.

1.2 Please ensure consistency in terminology. The abstract refers to “self-confidence” whereas the main text consistently uses “self-esteem.”

2. Methods Section

2.1 Procedure description lacks coherence and clarity.While the revised Procedure section includes several relevant elements—such as the survey period, platforms used, and eligibility criteria—the current version still reads as a list of disconnected points rather than a logically structured account of the data collection process. Rather than listing discrete elements, the procedure should be presented as a cohesive narrative, enabling readers to follow the actual implementation of the study step-by-step. To improve clarity and replicability, I recommend that the authors present this section as a coherent sequence of steps, clearly addressing:

1)Who: Who initiated and managed the survey distribution—was it the primary researchers, trained assistants, or both?

2)Where/How: Where and how was the survey shared (e.g., via WhatsApp, Instagram, LinkedIn)? Were any organizational or institutional platforms involved?

3)When: When was the survey accessible (precise start and end dates), and for how long?

4)What strategy: How was participation encouraged? For example, paid advertisement, direct messages, or snowball sampling. In particular, the statement “snowballing technique was encouraged” lacks detail.Please clarify how this was implemented—e.g., were participants explicitly asked to share the survey? Were any prompts, templates, or incentives provided? Was this process monitored or tracked?

5)Quality control: What steps were taken to ensure data integrity, such as screening for eligibility, controlling for duplicate responses, or ensuring informed consent?

2.2 The description of the SSQ still does not clearly distinguish whether Ochoa et al. (2015) is the scale developer or simply cited for psychometric validation. Please clarify the original source of the SSQ.

2.3 The placeholder “[ref]” remains in the statistical analysis section following the skewness/kurtosis normality statement. Please provide a proper reference to support the claim that skewness and kurtosis between -1 and +1 indicate normality.

3. Results

3.1 Table 2 presents informative bivariate comparisons; however, several aspects of its structure and presentation could be improved to enhance clarity and align with reporting standards:

(1)Structural clarity. The current presentation places group means and F/t statistics in the same row (e.g., under “Age” and “Education”), which may confuse readers as to whether the test statistics apply to individual subgroups or to the overall comparison. It is recommended to visually separate subgroup statistics from overall test results—for example, by indenting subgroup rows, using merged cells, or adding a distinct summary row for the F/t test.

(2)Effect size specification. The table includes effect size values but does not specify their type (e.g., Cohen’s d, eta-squared η²). For clarity, please identify the type of effect size used and provide interpretation guidance in a footnote (e.g., η² values of .01 = small, .06 = medium, .14 = large).

Addressing these points will improve the table’s clarity and help readers more accurately interpret your findings.You may wish to consult recently published tables in PLOS ONE for formatting consistency and reader accessibility.

3.2 The response suggests that stepwise regression results were included as “Supplementary Table 1,” but this table does not appear to be included in the revised manuscript or appendices. Please ensure it is attached and referenced properly.

4. Discussion

4.1 While the authors provide some general cultural context about Lebanon, the manuscript would benefit from a clearer discussion of how specific cultural elements—such as family dynamics, religious beliefs, or social expectations—might influence stigma and help-seeking behavior in this context. This need not be an exhaustive anthropological analysis, but even brief references to existing local studies, or examples of how such factors have been observed to shape attitudes in daily life or public discourse, would enhance the cultural grounding of the study.

4.2 The authors have included some general suggestions related to mental health education. To enhance the applied value of the study, it would be useful to briefly reflect on how these recommendations align with the realities of the Lebanese context. For example, are school-based awareness campaigns, university-level counseling workshops, or community-based mental health sessions feasible given current infrastructure or public attitudes? Even a brief reflection on what is currently possible or lacking would help to ground the study’s implications in the local context, without requiring detailed intervention plans.

5. Other Minor Issues

5.1 Spelling, grammar, and stylistic issues. During the review of your manuscript, I've identified several spelling errors and unclear expressions that should be corrected before final submission. Below are specific examples:

In the "Procedure" section:"...which are preferred common means of communication in Leabanon." → Please correct the country name spelling to "Lebanon"

In the description of the "Rosenberg Self-Esteem Scale":"It is a 4-point Linkert scale from 0 (strongly agree) to 3 (strongly disagree)."→ This should be corrected to "Likert scale"

In the "Procedure" section:"...shared via social media platforms of research personal such as WhatsApp..." → This should be revised to "research personnel"

In the paragraph following "Impact of Stigma on Self-Esteem":"Th negative self-perceptions and lower self-esteem associated with stigma..."→ This should be corrected to "The negative self-perceptions"

In the "Help-Seeking in relation to Self-Esteem" section: "University students with higher self-esteem seem to be queen of promptly recognizing..." → This expression is unclear and should be rephrased, perhaps as "keen on promptly" or another appropriate wording

In the discussion section related to stigma management theory:"...they would be more likely to seek support and around it." → This phrase is unclear; please revise to express the intended meaning more precisely

……

These revisions will enhance the professionalism and readability of your manuscript. I recommend conducting a thorough language review of the entire document before final submission to ensure no similar errors remain.

5.2 Causal language in a cross-sectional study. Some expressions (e.g., “X leads to Y”, “results in”) suggest causality. Given the cross-sectional nature of your study, please revise to use correlational phrasing (e.g., “X is associated with Y” or “may relate to Y”).

Overall, the manuscript presents promising findings and the revised version has addressed many core issues. If the authors can revise the remaining points as suggested above, the manuscript will be significantly strengthened. I therefore recommend Minor Revisions.

Reviewer #2: (No Response)

7. PLOS authors have the option to publish the peer review history of their article (what does this mean? ). If published, this will include your full peer review and any attached files.

**Do you want your identity to be public for this peer review?** For information about this choice, including consent withdrawal, please see our Privacy Policy .

Reviewer #1: No

Reviewer #2: No

---

## [Author Response · Author response to Decision Letter 2]

11 May 2025

Dear editor,

Thank you and the reviewers for your time investment to uplift the scientific rigor and display of our manuscript.

We have amended authors list to better reflect individual involvement in the review process. We have also added an expert to help address the culture-related major comments of the reviewer. We have secured authors alignment to move forward with this change.

We have subsequently carefully attended to comments and recommendations and are hereby submitting

• an updated version of the manuscript with highlighted tracked changes.

• a point-by-point reply in bold font to reviewers’ comments

We do look forward to your favorable input

On behalf of the team,

Dr Myriam EL Khoury-Malhame

Peer Review Comments

Dear Authors,

Thank you for your careful and thoughtful revisions to the manuscript. It is evident that substantial effort has been invested in improving the paper across multiple sections.

- Your incorporation of a theoretical framework

- Expansion of the literature review,

- Clarification of methodology,

- and inclusion of moderation plots have notably enhanced the overall clarity and contribution of the study.

Thanks a lot for the support throughout and for the constructive criticism.

After reviewing the revised manuscript, I find that the majority of the major concerns have been addressed. However, a few important issues remain unresolved or only partially addressed. Below is a summary of these points, organized by manuscript sections and marked by importance level.

1. Introduction / Background Section

1.1 While the narrative logic is coherent, the hypotheses remain embedded within the text. I recommend clearly enumerating them at the end of the Background section (e.g., H1, H2, etc.) for improved clarity and structure.

This has been done accordingly

1.2 Please ensure consistency in terminology. The abstract refers to “self-confidence” whereas the main text consistently uses “self-esteem.”

Terminology consistency has been checked.

2. Methods Section

2.1 Procedure description lacks coherence and clarity.While the revised Procedure section includes several relevant elements—such as the survey period, platforms used, and eligibility criteria—the current version still reads as a list of disconnected points rather than a logically structured account of the data collection process. Rather than listing discrete elements, the procedure should be presented as a cohesive narrative, enabling readers to follow the actual implementation of the study step-by-step.

Thank you for the mention, it is indeed important to us to improve narrative-like procedural reporting. As such we have actioned the below suggestions consistently.

To improve clarity and replicability, I recommend that the authors present this section as a coherent sequence of steps, clearly addressing:

- Who: Who initiated and managed the survey distribution—was it the primary researchers, trained assistants, or both? Both

- Where/How: Where and how was the survey shared (e.g., via WhatsApp, Instagram, LinkedIn)? Were any organizational or institutional platforms involved? Personal online platforms have been reported. No institutional platform was involved.

- When: When was the survey accessible (precise start and end dates), and for how long? This has been reported exactly

- What strategy: How was participation encouraged? For example, paid advertisement, direct messages, or snowball sampling. In particular, the statement “snowballing technique was encouraged” lacks detail.Please clarify how this was implemented—e.g., were participants explicitly asked to share the survey? Were any prompts, templates, or incentives provided? Was this process monitored or tracked? This has been clarified in text. Participants were indeed asked to share the link. The link was also sponsored via the social media platform to target Lebanese 18-40 years users. No incentives were provided, and the researchers had no way to track or monitor the snowballing process.

- Quality control: What steps were taken to ensure data integrity, such as screening for eligibility, controlling for duplicate responses, or ensuring informed consent? Informed consent was provided prior to filling the questionnaire by reading the ethical considerations and clicking an approve button. Duplicate responses were only tracked using IP addresses as no personal identifiers were requested to preserve anonymity and confidentiality. Eligibility was manually checked to only include participants over 18 residing in Lebanon.

2.2 The description of the SSQ still does not clearly distinguish whether Ochoa et al. (2015) is the scale developer or simply cited for psychometric validation. Please clarify the original source of the SSQ.

Thank you for the attention to detail. We have added the following “. [SSQ] is built on a shorter version of the original 29-item scale (Ritsher et al., 2003).”

2.3 The placeholder “[ref]” remains in the statistical analysis section following the skewness/kurtosis normality statement. Please provide a proper reference to support the claim that skewness and kurtosis between -1 and +1 indicate normality.

This has been amended and the reference has been added Field, A. (2013)

3. Results

3.1 Table 2 presents informative bivariate comparisons; however, several aspects of its structure and presentation could be improved to enhance clarity and align with reporting standards:

- Structural clarity. The current presentation places group means and F/t statistics in the same row (e.g., under “Age” and “Education”), which may confuse readers as to whether the test statistics apply to individual subgroups or to the overall comparison. It is recommended to visually separate subgroup statistics from overall test results—for example, by indenting subgroup rows, using merged cells, or adding a distinct summary row for the F/t test.

Sorry for the misunderstanding. The F/t statistics are in a distinct row than the means of the groups.

Table 2. Bivariate analysis of factors associated with attitude help seeking.

Variable Mean ± SD t / F df / df1, df2 p Effect size

Age (years) 8.88 3, 241 <.001 .100

18-19 2.84 ± .43

20-29 2.18 ± .73

30-39 2.94 ± .57

Sex -1.43 243 .155 .186

Male 2.17 ± .67

Female 2.31 ± .78

Education 10.82 3, 241 <.001 .119

High school 3.12 ± .64

Bachelor 2.15 ± .77

Master 2.37 ± .30

PhD 3.40 ± .42

Numbers in bold indicate significant p values.

Please let us know if you want us to make any modifications. Thank you.

- Effect size specification. The table includes effect size values but does not specify their type (e.g., Cohen’s d, eta-squared η2). For clarity, please identify the type of effect size used and provide interpretation guidance in a footnote (e.g., η2 values of .01 = small, .06 = medium, .14 = large).

We added this explanation as a footnote:

Effect size refers to Cohen’s d for sex (d values of .02 = small, .05 = medium, .08 = large) and eta-squared η2 (η2 values of .01 = small, .06 = medium, .14 = large) for age and education.

Addressing these points will improve the table’s clarity and help readers more accurately interpret your findings.You may wish to consult recently published tables in PLOS ONE for formatting consistency and reader accessibility.

Thank you for your suggestion. We made the changes accordingly.

3.2 The response suggests that stepwise regression results were included as “Supplementary Table 1,” but this table does not appear to be included in the revised manuscript or appendices. Please ensure it is attached and referenced properly.

We apologize for this oversight. We added the following to the results section:

Linear regressions and Moderation analyses

The results of two linear regressions taking the attitude help seeking as the dependent variable, psychological distress as the independent variable and self-esteem as the moderator and taking the attitude help seeking as the dependent variable, psychological distress as the independent variable and self-stigma as the moderator, can be found in supplementary tables 1 and 2 respectively.

4. Discussion

4.1 While the authors provide some general cultural context about Lebanon, the manuscript would benefit from a clearer discussion of how specific cultural elements—such as family dynamics, religious beliefs, or social expectations—might influence stigma and help-seeking behavior in this context. This need not be an exhaustive anthropological analysis, but even brief references to existing local studies, or examples of how such factors have been observed to shape attitudes in daily life or public discourse, would enhance the cultural grounding of the study.

Some of those elements were introduced in the background part and the discussion was centered to echo those cultural elements as recommended. This was mostly included in the conclusion segment.

4.2 The authors have included some general suggestions related to mental health education. To enhance the applied value of the study, it would be useful to briefly reflect on how these recommendations align with the realities of the Lebanese context. For example, are school-based awareness campaigns, university-level counseling workshops, or community-based mental health sessions feasible given current infrastructure or public attitudes? Even a brief reflection on what is currently possible or lacking would help to ground the study’s implications in the local context, without requiring detailed intervention plans.

This is an excellent point to gauge feasibility and cultural-relevance of those recommendations. A section has been added as some colleagues have already tested the importance of school-based campaigns and university workshops.

5. Other Minor Issues

5.1 Spelling, grammar, and stylistic issues. During the review of your manuscript, I've identified several spelling errors and unclear expressions that should be corrected before final submission. Below are specific examples:

- In the "Procedure" section:"...which are preferred common means of communication in Leabanon." → Please correct the country name spelling to "Lebanon"

- In the description of the "Rosenberg Self-Esteem Scale":"It is a 4-point Linkert scale from 0 (strongly agree) to 3 (strongly disagree)."→ This should be corrected to "Likert scale"

- In the "Procedure" section:"...shared via social media platforms of research personal such as WhatsApp..." → This should be revised to "research personnel"

- In the paragraph following "Impact of Stigma on Self-Esteem":"Th negative self-perceptions and lower self-esteem associated with stigma..."→ This should be corrected to "The negative self-perceptions"

- In the "Help-Seeking in relation to Self-Esteem" section: "University students with higher self-esteem seem to be queen of promptly recognizing..." → This expression is unclear and should be rephrased, perhaps as "keen on promptly" or another appropriate wording

- In the discussion section related to stigma management theory:"...they would be more likely to seek support and around it." → This phrase is unclear; please revise to express the intended meaning more precisely .

These revisions will enhance the professionalism and readability of your manuscript. I recommend conducting a thorough language review of the entire document before final submission to ensure no similar errors remain.

We apologize for the one-too-many mistakes and agree it undermines the quality of the submission. We have reviewed the manuscript thoroughly and hope to have increased professionalism and readability.

5.2 Causal language in a cross-sectional study. Some expressions (e.g., “X leads to Y”, “results in”) suggest causality. Given the cross-sectional nature of your study, please revise to use correlational phrasing (e.g., “X is associated with Y” or “may relate to Y”).

Agreed and done.

Overall, the manuscript presents promising findings and the revised version has addressed many core issues. If the authors can revise the remaining points as suggested above, the manuscript will be significantly strengthened. I therefore recommend Minor Revisions.

We have recruited a public health expert to upend the cultural aspect in relation to the manuscript as well as to highlight actionable evidence-based recommendations. We have also carefully attended to the minor revisions and do look forward for the publication of this paper.

---

## [Decision Letter · Decision Letter 2]

27 May 2025

PONE-D-24-53040R2Cultural Stigma, Psychological Distress and Help-Seeking: Moderating role of Self-esteem and Self-stigmaPLOS ONE

Dear Dr. El Khoury Malhame,

Thank you for submitting your manuscript to PLOS ONE. After careful consideration, we feel that it has merit but does not fully meet PLOS ONE’s publication criteria as it currently stands. Therefore, we invite you to submit a revised version of the manuscript that addresses the points raised during the review process.

We look forward to receiving your revised manuscript.

Kind regards,

Lakshminarayana Chekuri, MD, PhD

Academic Editor

PLOS ONE

Additional Editor Comments :

Thank you for your scholarly contribution. I'd also like to thank the authors for choosing PLOS ONE to publish your findings from this study. Comments from reviewer are provided below. Please review these comments and I suggest address them and resubmit your manuscript. Your timely response would help this study be published and will make it accessible to interested readers across the world. I look forward to reviewing your revised manuscript. I wish you good luck with your future endeavors.

Reviewers' comments:

Reviewer's Responses to Questions

**Comments to the Author**

1. If the authors have adequately addressed your comments raised in a previous round of review and you feel that this manuscript is now acceptable for publication, you may indicate that here to bypass the “Comments to the Author” section, enter your conflict of interest statement in the “Confidential to Editor” section, and submit your "Accept" recommendation.

Reviewer #1: (No Response)

2. Is the manuscript technically sound, and do the data support the conclusions?

Reviewer #1: Partly

3. Has the statistical analysis been performed appropriately and rigorously? 

Reviewer #1: No

4. Have the authors made all data underlying the findings in their manuscript fully available?

Reviewer #1: Yes

5. Is the manuscript presented in an intelligible fashion and written in standard English?

Reviewer #1: Yes

6. Review Comments to the Author

Reviewer #1: Peer Review Report

Manuscript Title: Cultural Stigma, Psychological Distress and Help-Seeking: Moderating role of Self-esteem and Self-stigma in Lebanon

Date of Review: May 23, 2025

Reviewer Comments:

First, I would like to thank the author team for their response to previous review comments and the revisions made. It is evident that efforts have been made to improve the quality of the manuscript, such as deepening the background, refining the description of the methods section, and correcting some minor issues. These positive changes are acknowledged.

However, upon a further detailed review and comparison of Revision 2 - Second revised manuscript with the previous Revision 1- First revised manuscript and the Original Submission-Original manuscript, I have noted some critical issues that were perhaps not fully uncovered in previous reviews, or whose deeper implications merit further emphasis. If these issues are not fundamentally addressed, they will challenge the reliability of the study's conclusions and its overall academic rigor.

This study explores the complex relationships between cultural stigma, psychological distress, self-esteem, self-stigma, and help-seeking behaviors among young adults in Lebanon, a topic of significant practical and academic value. However, the current presentation of data and the articulation of some core research elements make it difficult to accurately assess the authenticity and validity of the research findings.

Major Issues

Q1:Regarding the Consistency between the Reported Sample Size and the Actual Basis for Data Analysis

The manuscript, across different versions and sections (e.g., the “Participants” subsection of the Methods and the title of Table 1), has revised the final sample size from N=245 to N=243, stating the removal of 2 over-age participants and 14 incomplete responses. However, a careful examination of the data details within the manuscript reveals some concerning signs. For instance, in Table 1, the sum of N-values for “Sex” and “Education” categories still amounts to 245 participants, and their percentage calculations also appear to be based on N=245. More critically, the degrees of freedom (df) for t-tests and ANOVAs in Table 2 consistently point to an analytical sample base of N=245 (e.g., the t-test for “Sex” has df=243, implying N=245; the ANOVA for “Education” has df2=241, which for 4 groups implies N=245). Furthermore, some textual descriptions in the abstract and results section still refer to 245 participants.

Concurrently, the description of the sample exclusion process in the Methods section (“The final sample consisted of 243 participants...14...excluded...2...removed from subsequent analyses.”) could be clearer regarding how the final analytical sample of 243 was precisely derived from an initial pool of participants. The current phrasing (e.g., “removed from subsequent analyses”) and the lack of explicit mention of the initial total number of individuals before these exclusions might lead to reader queries about the completeness and precision of the screening process.

If the actual statistical analyses were conducted on a sample size inconsistent with the finally reported one, this would constitute a fundamental issue of data accuracy, directly affecting the validity of all descriptive and inferential statistical results. It is strongly recommended that the authors: Clearly define a unified, final sample size for all analyses. Based on this sample size, thoroughly re-clean and re-analyze all original data. Subsequently, comprehensively update all relevant figures and statements in the manuscript (including N-values, percentages, means, standard deviations, degrees of freedom, test statistics, p-values, regression coefficients, confidence intervals, etc., in text, tables, and figures) to ensure complete data consistency and accuracy throughout. Please detail, clearly and unambiguously, each step of the participant screening process from initial recruitment to final inclusion in the analysis, along with corresponding participant numbers, in the Methods section.

Q2:Regarding the Appropriateness of the Model Fit Index in Moderation Analyses (Table 4)

I noted that in Table 4 (Moderation analyses results) of the current manuscript, the authors report “Nagelkerke R²” as the model fit index (e.g., “Model 1: Self-esteem as the moderator (Nagelkerke R² = 0.437; F = 18.15)” ).

Typically, Nagelkerke R² (or Pseudo R-squared) is a common metric for assessing the goodness-of-fit of Logistic Regression models, where the dependent variable is categorical. However, judging from the description of the dependent variable “attitude help seeking” in the manuscript (e.g., mean and standard deviation of 2.26 ± .74 in Table 1, and its description as “considered normally distributed” in the statistical analysis section ), it appears to be treated as a continuous variable. The PROCESS MACRO used for moderation analysis, when dealing with continuous dependent variables, is typically based on linear regression models. The Beta coefficients, t-values, and p-values reported in Table 4 also align with the characteristic output of linear regression.

If a linear regression model was indeed used, reporting Nagelkerke R² could mislead readers in their judgment of the model's explanatory power and would reflect imprecision in statistical methodology selection and reporting. It is recommended that the authors: Carefully verify the specific model type used in their moderation analyses and the nature of its dependent variable. If it is indeed a linear regression model for a continuous dependent variable, the Nagelkerke R² reported in Table 4 should be replaced with the standard R² (coefficient of determination) or Adjusted R², and ensure the F-value reporting is consistent with the chosen R² type and model.

Q3:Regarding the Clarity and Accuracy of Research Hypothesis Formulation

The research hypotheses H1, H2, and H3, as listed in the current manuscript, could be further optimized to enhance their normative value and clarity as scientific hypotheses.

For instance, H1’s phrasing “stigma contributes to increased psychological distress” carries strong causal implications, which might be overly assertive for a cross-sectional study where correlational language is often more cautious.

H2’s phrasing “We mostly posit that... self-esteem, could moderate...” appears somewhat tentative. More critically, this hypothesis (self-esteem moderating the relationship between stigma and self-stigma) does not seem to align perfectly with the moderation analysis actually conducted and reported in the manuscript (Table 4 Model 1 tests self-esteem as a moderator of the relationship between psychological distress and help-seeking attitude).

H3’s formulation “This could be accounted for because...” reads more like an explanation or an inference for a phenomenon rather than a directly testable predictive statement.

The statement of research hypotheses should strive for clarity, precision, testability, and strict alignment with the research design and the actual data analysis methods employed.It is suggested that the authors rephrase the hypotheses with more direct and affirmative language. For cross-sectional studies, prioritize using terms that describe relationships between variables. Crucially, ensure that H2 aligns with the moderation model actually analyzed or that analyses testing the original H2 are supplemented. H3 might be reformulated as a more direct relational prediction or its logic integrated into the theoretical exposition of other hypotheses.

Q4:Regarding the Reporting and Interpretation of Psychological Distress (Kessler K6) Scores

The “Measures”section of the manuscript states that the Kessler scale total score ranges from 0-24, with a score ≥13 indicating higher distress. However, Table 1 reports the sample's mean psychological distress score as 3.06 (SD=0.93). If this is indeed a total score on the 0-24 scale, it suggests a very low average level of distress for a sample described as “diagnosed with at least one mental illness.” The x-axis range for psychological distress in Figures 1 and 2 (approximately 2.13 to 3.99) also seems to reflect this lower score range, which adds to the interpretative challenge.

The authors urgently need to clarify in the “Measures” section and in the notes for Table 1 whether the reported K6 psychological distress score is a total sum score or a mean item score. Based on this clarification, it is imperative to re-evaluate whether the mean score in Table 1 is consistent with the sample characteristics and ensure that all subsequent delimitations and discussions of “high/low psychological distress” in the manuscript align with this clarification.

Minor Issues

Q1:Regarding the Pervasiveness of Language Strongly Implying Causality

Despite the authors' previous indication (in response to R2 review point 5.2) that causal language would be revised, a thorough reading of the current manuscript’s Abstract, Background, Discussion, and even Conclusion sections still reveals numerous instances of phrases with strong causal implications (e.g., “leads to”, “results in”, “causes”, “worsens”, “degrades”, “shapes”, “contributes to”, “impacts”, ......). These terms, when describing relationships between variables, carry varying degrees of causal directionality. For a cross-sectional study design, which has inherent limitations in inferring causality, such expressions can easily lead to an over-interpretation of the findings.

The authors should undertake a systematic review of the entire text to revise all statements with strong causal implications, replacing them with phrasing more appropriate for describing associations or correlations found in cross-sectional research (e.g., consider using “is associated with,” “is related to,” “correlates with,” “is linked to,” “may correspond with,” “is observed alongside”). This revision should be consistently applied throughout the manuscript to ensure the rigor of the research report and the accuracy of its statements.

Q2:Accuracy of Percentage Summation in Tables

In Table 1, the sum of percentages for “Age” categories (99.2%) and “Education” categories (100.4%) do not precisely equal 100%. The sum of percentages for types of mental illnesses in the Results text exceeds 100% (110.5%).

Please carefully check the calculation and rounding of percentages in Table 1 for each category to ensure their sum is within a reasonable rounding error of 100%. For the types of mental illness, if multiple diagnoses are allowed, leading to a sum >100%, this should be explicitly stated in the text to avoid reader misinterpretation.

Q3:Interpretation of Self-Stigma Scores in the Discussion

The Self-Stigma Questionnaire (SSQ) is scored such that “higher scores indicating lower levels of self-stigma”.

When discussing the moderation effects in Table 5 related to low SSQ scores (e.g., 1.27), it is advisable to explain more clearly and consistently that this represents a state of “high self-stigma” (at the construct level) to help readers accurately grasp the direction and meaning of the moderation effect.

Q4:Accuracy of Statistical Table Details

The degrees of freedom (df1=3) for the Age ANOVA in Table 2 appear inconsistent with the three age groups listed in Table 1 (if three groups were actually compared, df1 should be 2).

Please verify the actual number of groups used in the ANOVA for Age and ensure that the reported degrees of freedom in Table 2 accurately reflect the analytical design.

Overall Conclusion and Recommendations:

Considering the severity of the major issues identified, particularly regarding the fundamental discrepancy between the data analysis basis and the reported sample size, the inappropriate reporting of the R² statistic, the problematic formulation and mismatch of research hypotheses, critical questions about the interpretation of psychological distress scores, and the pervasive use of causal language, I recommend that this manuscript requires Major Revision.

Only after these critical data and core research element issues are thoroughly addressed, and all analyses are based on a clearly, consistently, and accurately reported sample size using appropriate statistical reporting methods, can the scientific merit and reliability of this study's conclusions be more accurately assessed. The current state prevents the reviewer from fully trusting the presented results and the study’s logical and methodological soundness.

7. PLOS authors have the option to publish the peer review history of their article (what does this mean? ). If published, this will include your full peer review and any attached files.

**Do you want your identity to be public for this peer review?** For information about this choice, including consent withdrawal, please see our Privacy Policy .

Reviewer #1: No

---

## [Author Response · Author response to Decision Letter 3]

4 Jun 2025

Dear Editor

We thank you for your compassionate approach in delivering feedback and incentivizing the team to keep polishing the manuscript until it reaches journal standards.

We also do acknowledge the commendable efforts of the reviewers as well as their remarkable attention to details.

We do hope this final edited version addresses all raised comments. For ease of tracking, we have hereby provided our answers point-by-point in bold and highlighted changes in the manuscript accordingly upon request.

We look forward to making this accessible to readers across the world as you so genuinely suggest

Thank you again

Dr Myriam eL Khoury-Malhame

Additional Editor Comments :

Thank you for your scholarly contribution. I'd also like to thank the authors for choosing PLOS ONE to publish your findings from this study. Comments from reviewer are provided below. Please review these comments and I suggest address them and resubmit your manuscript.

Your timely response would help this study be published and will make it accessible to interested readers across the world. I look forward to reviewing your revised manuscript. I wish you good luck with your future endeavors.

Reviewer #1: Peer Review Report

Manuscript Title: Cultural Stigma, Psychological Distress and Help-Seeking: Moderating role of Self-esteem and Self-stigma in Lebanon

Date of Review: May 23, 2025

Reviewer Comments:

First, I would like to thank the author team for their response to previous review comments and the revisions made. It is evident that efforts have been made to improve the quality of the manuscript, such as deepening the background, refining the description of the methods section, and correcting some minor issues. These positive changes are acknowledged.

Much appreciated comments. Thank you for taking the time to accompany those modifications.

However, upon a further detailed review and comparison of Revision 2 - Second revised manuscript with the previous Revision 1- First revised manuscript and the Original Submission-Original manuscript, I have noted some critical issues that were perhaps not fully uncovered in previous reviews, or whose deeper implications merit further emphasis. If these issues are not fundamentally addressed, they will challenge the reliability of the study's conclusions and its overall academic rigor.

This study explores the complex relationships between cultural stigma, psychological distress, self-esteem, self-stigma, and help-seeking behaviors among young adults in Lebanon, a topic of significant practical and academic value. However, the current presentation of data and the articulation of some core research elements make it difficult to accurately assess the authenticity and validity of the research findings.

Thank you for your comments, we have addressed them to the best of our knowledge to maintain authenticity and validity of the research arm of the project all while providing the practical clinical value and therapeutic implications.

Major Issues

Q1:Regarding the Consistency between the Reported Sample Size and the Actual Basis for Data Analysis

The manuscript, across different versions and sections (e.g., the “Participants” subsection of the Methods and the title of Table 1), has revised the final sample size from N=245 to N=243, stating the removal of 2 over-age participants and 14 incomplete responses. However, a careful examination of the data details within the manuscript reveals some concerning signs. For instance, in Table 1, the sum of N-values for “Sex” and “Education” categories still amounts to 245 participants, and their percentage calculations also appear to be based on N=245.

Thank you for the highlight. We went back to the raw data and checked numbers closely.

The final sample size was indeed 245 AFTER removing the 14 incomplete responses with more than 30% missing values (initially 259 participants). The 2 participants who were 40 were actually kept in the analyses as the age bracket allowed inclusion. All previous analyses were run at submission (and re-run at this point) on the basis of 245. The reporting however was confusing.

We do massively apologize for the typos while exporting the values from the excel and SPSS sheets.

More critically, the degrees of freedom (df) for t-tests and ANOVAs in Table 2 consistently point to an analytical sample base of N=245 (e.g., the t-test for “Sex” has df=243, implying N=245; the ANOVA for “Education” has df2=241, which for 4 groups implies N=245). Furthermore, some textual descriptions in the abstract and results section still refer to 245 participants.

The analytical sample base is indeed 245 and we have run all analyses again to make sure all data and numbers are relevant to this initial sample size. We apologize once again for this typo mistake, that has inadvertently caused, as you righteously point, major inconsistencies.

Concurrently, the description of the sample exclusion process in the Methods section (“The final sample consisted of 243 participants...14...excluded...2...removed from subsequent analyses.”) could be clearer regarding how the final analytical sample of 243 was precisely derived from an initial pool of participants. The current phrasing (e.g., “removed from subsequent analyses”) and the lack of explicit mention of the initial total number of individuals before these exclusions might lead to reader queries about the completeness and precision of the screening process.

This has been clarified in the manuscript to avoid such confusions. Initial sample size of 259 was reported. The 2 participants that were removed were actually 40yrs sharp so in subsequent analyses we decided to keep them as they fit into the initial age range. The intent was to explain that we removed the 14 and kept the other 2. We have properly reformulated the section accordingly.

If the actual statistical analyses were conducted on a sample size inconsistent with the finally reported one, this would constitute a fundamental issue of data accuracy, directly affecting the validity of all descriptive and inferential statistical results. It is strongly recommended that the authors: Clearly define a unified, final sample size for all analyses. Based on this sample size, thoroughly re-clean and re-analyze all original data. Subsequently, comprehensively update all relevant figures and statements in the manuscript (including N-values, percentages, means, standard deviations, degrees of freedom, test statistics, p-values, regression coefficients, confidence intervals, etc., in text, tables, and figures) to ensure complete data consistency and accuracy throughout. Please detail, clearly and unambiguously, each step of the participant screening process from initial recruitment to final inclusion in the analysis, along with corresponding participant numbers, in the Methods section.

The final sample included in the analysis is 245. This was corrected throughout the manuscript. To avoid confusion we have clarified that 14 questionnaires with more than 30% missing data were removed.

Q2:Regarding the Appropriateness of the Model Fit Index in Moderation Analyses (Table 4)

I noted that in Table 4 (Moderation analyses results) of the current manuscript, the authors report “Nagelkerke R²” as the model fit index (e.g., “Model 1: Self-esteem as the moderator (Nagelkerke R² = 0.437; F = 18.15)” ).

Typically, Nagelkerke R² (or Pseudo R-squared) is a common metric for assessing the goodness-of-fit of Logistic Regression models, where the dependent variable is categorical. However, judging from the description of the dependent variable “attitude help seeking” in the manuscript (e.g., mean and standard deviation of 2.26 ± .74 in Table 1, and its description as “considered normally distributed” in the statistical analysis section ), it appears to be treated as a continuous variable. The PROCESS MACRO used for moderation analysis, when dealing with continuous dependent variables, is typically based on linear regression models. The Beta coefficients, t-values, and p-values reported in Table 4 also align with the characteristic output of linear regression.

If a linear regression model was indeed used, reporting Nagelkerke R² could mislead readers in their judgment of the model's explanatory power and would reflect imprecision in statistical methodology selection and reporting. It is recommended that the authors: Carefully verify the specific model type used in their moderation analyses and the nature of its dependent variable. If it is indeed a linear regression model for a continuous dependent variable, the Nagelkerke R² reported in Table 4 should be replaced with the standard R² (coefficient of determination) or Adjusted R², and ensure the F-value reporting is consistent with the chosen R² type and model.

Thank you for the comment. We replaced the term “Nagelkerke” by “Standard” and made sure that the F value is correct for the chosen R2 type and model.

Q3:Regarding the Clarity and Accuracy of Research Hypothesis Formulation

The research hypotheses H1, H2, and H3, as listed in the current manuscript, could be further optimized to enhance their normative value and clarity as scientific hypotheses.

For instance, H1’s phrasing “stigma contributes to increased psychological distress” carries strong causal implications, which might be overly assertive for a cross-sectional study where correlational language is often more cautious.

To enhance the normative value and clarity of the research hypotheses, we have rephrased the hypotheses

H1: Higher levels of perceived stigma are expected to be associated with increased psychological distress.

H2’s phrasing “We mostly posit that... self-esteem, could moderate...” appears somewhat tentative. More critically, this hypothesis (self-esteem moderating the relationship between stigma and self-stigma) does not seem to align perfectly with the moderation analysis actually conducted and reported in the manuscript (Table 4 Model 1 tests self-esteem as a moderator of the relationship between psychological distress and help-seeking attitude).

H2: Self-esteem is hypothesized to moderate the relationship between psychological distress and help-seeking attitudes, such that individuals with higher self-esteem may show more favorable help-seeking attitudes even when experiencing psychological distress.

H3’s formulation “This could be accounted for because...” reads more like an explanation or an inference for a phenomenon rather than a directly testable predictive statement.

The statement of research hypotheses should strive for clarity, precision, testability, and strict alignment with the research design and the actual data analysis methods employed.It is suggested that the authors rephrase the hypotheses with more direct and affirmative language. For cross-sectional studies, prioritize using terms that describe relationships between variables. Crucially, ensure that H2 aligns with the moderation model actually analyzed or that analyses testing the original H2 are supplemented. H3 might be reformulated as a more direct relational prediction or its logic integrated into the theoretical exposition of other hypotheses.

H3: Greater stigma is expected to be associated with less favorable help-seeking attitudes.

Q4: Regarding the Reporting and Interpretation of Psychological Distress (Kessler K6) Scores

The “Measures”section of the manuscript states that the Kessler scale total score ranges from 0-24, with a score ≥13 indicating higher distress. However, Table 1 reports the sample's mean psychological distress score as 3.06 (SD=0.93). If this is indeed a total score on the 0-24 scale, it suggests a very low average level of distress for a sample described as “diagnosed with at least one mental illness.” The x-axis range for psychological distress in Figures 1 and 2 (approximately 2.13 to 3.99) also seems to reflect this lower score range, which adds to the interpretative challenge.

The authors urgently need to clarify in the “Measures” section and in the notes for Table 1 whether the reported K6 psychological distress score is a total sum score or a mean item score. Based on this clarification, it is imperative to re-evaluate whether the mean score in Table 1 is consistent with the sample characteristics and ensure that all subsequent delimitations and discussions of “high/low psychological distress” in the manuscript align with this clarification.

We deleted the previous sentences about the total score and added the following:

The mean total score was computed, with its values varying between 0 and 4.

Minor Issues

Q1:Regarding the Pervasiveness of Language Strongly Implying Causality

Despite the authors' previous indication (in response to R2 review point 5.2) that causal language would be revised, a thorough reading of the current manuscript’s Abstract, Background, Discussion, and even Conclusion sections still reveals numerous instances of phrases with strong causal implications (e.g., “leads to”, “results in”, “causes”, “worsens”, “degrades”, “shapes”, “contributes to”, “impacts”, ......). These terms, when describing relationships between variables, carry varying degrees of causal directionality. For a cross-sectional study design, which has inherent limitations in inferring causality, such expressions can easily lead to an over-interpretation of the findings.

The authors should undertake a systematic review of the entire text to revise all statements with strong causal implications, replacing them with phrasing more appropriate for describing associations or correlations found in cross-sectional research (e.g., consider using “is associated with,” “is related to,” “correlates with,” “is linked to,” “may correspond with,” “is observed alongside”). This revision should be consistently applied throughout the manuscript to ensure the rigor of the research report and the accuracy of its statements.

The suggested terms have been used accordingly and the manuscript has been thoroughly revised to avoid causal implications at best when unjustified.

Q2: Accuracy of Percentage Summation in Tables

In Table 1, the sum of percentages for “Age” categories (99.2%) and “Education” categories (100.4%) do not precisely equal 100%. The sum of percentages for types of mental illnesses in the Results text exceeds 100% (110.5%).

Please carefully check the calculation and rounding of percentages in Table 1 for each category to ensure their sum is within a reasonable rounding error of 100%. For the types of mental illness, if multiple diagnoses are allowed, leading to a sum >100%, this should be explicitly stated in the text to avoid reader misinterpretation.

This has been corrected now. We apologize for this oversight.

Q3:Interpretation of Self-Stigma Scores in the Discussion

The Self-Stigma Questionnaire (SSQ) is scored such that “higher scores indicating lower levels of self-stigma”.

When discussing the moderation effects in Table 5 related to low SSQ scores (e.g., 1.27), it is advisable to explain more clearly and consistently that this represents a state of “high self-stigma” (at the construct level) to help readers accurately grasp the direction and meaning of the moderation effect.

We made the changes as requested as follows:

This suggests that individuals with varying levels of self-stigma (reflected by SSQ scores, where lower scores indicate higher self-stigma) may differ in their likelihood to seek help when experiencing psychological distress. At low (high self-stigma) (Beta = -.28; p = .001) and moderate (moderate self-stigma) (Beta = -.17; p = .011) levels of self-stigma, higher psychological distress was significantly associated with lower help-seeking attitude (Table 5, Model 2).

Q4:Accuracy of Statistical Table Details

The degrees of freedom (df1=3) for the Age ANOVA in Table 2 appear inconsistent with the three age groups listed in Table 1 (if three groups were actually compared, df1 should be 2).

Please verify the actual number of groups used in the ANOVA for Age and ensure that the reported degrees of freedom in Table 2 accurately reflect the analytical design.

We had 4 age groups. This has been corrected now. Please accept our apologies for the inattention.

Overall Conclusion and Recommendations:

Conside

---

## [Decision Letter · Decision Letter 3]

9 Jun 2025

PONE-D-24-53040R3Cultural Stigma, Psychological Distress and Help-Seeking:

Moderating role of Self-esteem and Self-stigmaPLOS ONE

Dear Dr. El Khoury Malhame,

Thank you for submitting your manuscript to PLOS ONE. After careful consideration, we feel that it has merit but does not fully meet PLOS ONE’s publication criteria as it currently stands. Therefore, we invite you to submit a revised version of the manuscript that addresses the points raised during the review process.

We look forward to receiving your revised manuscript.

Kind regards,

Lakshminarayana Chekuri, MD, PhD

Academic Editor

PLOS ONE

Journal Requirements:

Additional Editor Comments :

Thank you for your scholarly contribution. I'd also like to thank the authors for choosing PLOS ONE to publish your findings from this study. Comments from reviewers are provided below. Please review these comments and I suggest address them and resubmit your manuscript. Your timely response would help this study be published and will make it accessible to interested readers across the world. I look forward to reviewing your revised manuscript. I wish you good luck with your future endeavors.

Reviewers' comments:

Reviewer's Responses to Questions

**Comments to the Author**

1. If the authors have adequately addressed your comments raised in a previous round of review and you feel that this manuscript is now acceptable for publication, you may indicate that here to bypass the “Comments to the Author” section, enter your conflict of interest statement in the “Confidential to Editor” section, and submit your "Accept" recommendation.

Reviewer #1: (No Response)

2. Is the manuscript technically sound, and do the data support the conclusions?

Reviewer #1: Yes

3. Has the statistical analysis been performed appropriately and rigorously? 

Reviewer #1: Yes

4. Have the authors made all data underlying the findings in their manuscript fully available?

Reviewer #1: Yes

5. Is the manuscript presented in an intelligible fashion and written in standard English?

Reviewer #1: Yes

6. Review Comments to the Author

Reviewer #1: Manuscript Title: Cultural Stigma, Psychological Distress and Help-Seeking: Moderating role of Self-esteem and Self-stigma in Lebanon

1. General Comments

The authors have responded to previous reviewer comments in this revision, and the manuscript has improved in terms of data consistency and the alignment between hypotheses and analyses. The study’s topic, concerning mental health stigma and its correlates in the Lebanese cultural context, is of research value.

However, for the manuscript to meet the standard for publication, several issues still require attention. Revisions are recommended to further enhance the manuscript’s scientific rigor, clarity, and accuracy in its interpretation, methodological reporting, and language.

2. Specific Issues for Correction

2.1. Rigor in Interpretation and Language

Correction of Causal Language:The study employs a cross-sectional design, which does not support causal conclusions. However, the text frequently uses language that implies causality (e.g., “leading to” in the Abstract; “deters” and “exacerbating” in the Discussion), which constitutes an over-interpretation of the findings. It is recommended that the authors systematically revise the manuscript to replace such terms with more precise, associative language (e.g., “is associated with,” “is related to”) to ensure the conclusions are rigorously supported by the data.

2.2. Accuracy in Methods and Results Reporting

2.2.1 In the ‘Participants’ subsection of the Methods, the sentence “The number of participant responses that were excluded from the analysis was 14 due to incomplete survey responses.” is repeated verbatim. One instance should be removed.

2.2.2 In the Results section, the sum of the percentages for different mental illness diagnoses exceeds 100%. This suggests participant comorbidity, but this is not explicitly stated. A brief explanatory note should be added to clarify this for the reader.

2.3. Language and Formatting Corrections

2.3.1 Spelling Errors:

In the ‘Focus on the Lebanese Context’ subsection, “loosers” should be corrected to “losers”.

In the ‘Limitations and Future Research’ subsection, “ofour” should be corrected to “of our”.

2.3.2 Grammar and Punctuation:

In the ‘Participants’ subsection, there is a stray space before the period in “researchers .”.

In the description of the Self-Stigma Questionnaire SSQ, the phrase “high level of internal consistency, and Cronbach's alpha coefficients ranging from 0.75 to 0.901.” is a sentence fragment and should be corrected.

2.3.3 Phrasing and Word Choice:

In the Discussion, “complexified” is non-standard and should be replaced with “complicated”.

In the Statistical Analysis section, “adjusted over the following covariates” is unidiomatic. The standard preposition is “for.”

The phrase “The student t” is informal. It should be revised to the more standard “A Student's t-test.”

3. Final Recommendation

The core research presented in the manuscript is sound. However, the manuscript requires a thorough proofreading to address a number of basic errors before it can be considered ready for publication. I strongly recommend that the authors conduct a comprehensive review of the entire text to correct all errors in grammar, spelling, punctuation, and capitalization, including the specific examples provided in this report.

Once these essential corrections have been made, the manuscript will represent a valuable contribution to the literature. Therefore, my recommendation is as follows: Minor Revision.

7. PLOS authors have the option to publish the peer review history of their article (what does this mean? ). If published, this will include your full peer review and any attached files.

**Do you want your identity to be public for this peer review?** For information about this choice, including consent withdrawal, please see our Privacy Policy .

Reviewer #1: No

---

## [Author Response · Author response to Decision Letter 4]

12 Jun 2025

Dearest academic editor

Once again, my team and I are grateful for the timely constructive follow-up

Thank you for your patience as we finalize the uplift of our paper. We have hereby addressed the reviewer comments in bold and have highlighted changes in the manuscript accordingly.

We do hope these changes make the manuscript suitable for sharing

Best

Myriam EL Khoury-Malhame

CC: lchekuri@kansascity.edu, lchekuri@freemanhealth.com

PONE-D-24-53040R3

Cultural Stigma, Psychological Distress and Help-Seeking:

Moderating role of Self-esteem and Self-stigma

PLOS ONE

Dear Dr. El Khoury Malhame,

Thank you for submitting your manuscript to PLOS ONE. After careful consideration, we feel that it has merit but does not fully meet PLOS ONE’s publication criteria as it currently stands. Therefore, we invite you to submit a revised version of the manuscript that addresses the points raised during the review process.

Manuscript Title: Cultural Stigma, Psychological Distress and Help-Seeking:

Moderating role of Self-esteem and Self-stigma in Lebanon

1. General Comments

The authors have responded to previous reviewer comments in this revision, and the

manuscript has improved in terms of data consistency and the alignment between

hypotheses and analyses. The study’s topic, concerning mental health stigma and its

correlates in the Lebanese cultural context, is of research value.

Thank you, much appreciated

However, for the manuscript to meet the standard for publication, several issues still

require attention. Revisions are recommended to further enhance the manuscript’s

scientific rigor, clarity, and accuracy in its interpretation, methodological reporting,

and language.

This has been addressed accordingly.

2. Specific Issues for Correction

2.1. Rigor in Interpretation and Language

Correction of Causal Language:The study employs a cross-sectional design, which

does not support causal conclusions. However, the text frequently uses language that

implies causality (e.g., “leading to” in the Abstract; “deters” and “exacerbating” in the

Discussion), which constitutes an over-interpretation of the findings. It is

recommended that the authors systematically revise the manuscript to replace such

terms with more precise, associative language (e.g., “is associated with,” “is related

to”) to ensure the conclusions are rigorously supported by the data.

Absolutely on point comment. The team had previously edited the Introduction and has indeed shown less consistency in revisiting abstract and Discussion. A thorough review addressing the above has been done.

2.2. Accuracy in Methods and Results Reporting

2.2.1 In the ‘Participants’ subsection of the Methods, the sentence “The number of

participant responses that were excluded from the analysis was 14 due to incomplete

survey responses.” is repeated verbatim. One instance should be removed.

Done. Apologies for the shortsighted repetition.

2.2.2 In the Results section, the sum of the percentages for different mental illness

diagnoses exceeds 100%. This suggests participant comorbidity, but this is not

explicitly stated. A brief explanatory note should be added to clarify this for the

reader.

The explanation of comorbidity has been added accordingly. “Some participants reported comorbid diagnoses i.e., they documented being diagnosed with more than one mental disorder”

2.3. Language and Formatting Corrections

2.3.1 Spelling Errors:

In the ‘Focus on the Lebanese Context’ subsection, “loosers” should be corrected

to “losers”.

Done.

In the ‘Limitations and Future Research’ subsection, “ofour” should be corrected

to “of our”.

We couldn’t find this iteration

2.3.2 Grammar and Punctuation:

In the ‘Participants’ subsection, there is a stray space before the period in

“researchers .”.

Done at the end of the procedure section, the period was in bold. The format has been corrected.

In the description of the Self-Stigma Questionnaire SSQ, the phrase “high level of

internal consistency, and Cronbach's alpha coefficients ranging from 0.75 to

0.901.” is a sentence fragment and should be corrected.

Done. It now reads “The scale systematically shows a high level…”

2.3.3 Phrasing and Word Choice:

In the Discussion, “complexified” is non-standard and should be replaced with

“complicated”.

Done

In the Statistical Analysis section, “adjusted over the following covariates” is

unidiomatic. The standard preposition is “for.”

Done for the 2 instances

The phrase “The student t” is informal. It should be revised to the more standard

“A Student's t-test.”

Absolutely. Apologies for the informality. This has been corrected

3. Final Recommendation

The core research presented in the manuscript is sound. However, the manuscript

requires a thorough proofreading to address a number of basic errors before it can be

considered ready for publication. I strongly recommend that the authors conduct a

comprehensive review of the entire text to correct all errors in grammar, spelling,

punctuation, and capitalization, including the specific examples provided in this

report.

Once these essential corrections have been made, the manuscript will represent a

valuable contribution to the literature. Therefore, my recommendation is as follows:

Minor Revision.

Amazing closeup attention to details and much needed recommendations. The team acknowledges the vast knowledge and the generosity of the reviewer with his/her time and energy.

---

## [Decision Letter · Decision Letter 4]

19 Jun 2025

PONE-D-24-53040R4Cultural Stigma, Psychological Distress and Help-Seeking:

Moderating role of Self-esteem and Self-stigma in LebanonPLOS ONE

Dear Dr. El Khoury Malhame,

Thank you for submitting your manuscript to PLOS ONE. After careful consideration, we feel that it has merit but does not fully meet PLOS ONE’s publication criteria as it currently stands. Therefore, we invite you to submit a revised version of the manuscript that addresses the points raised during the review process.

We look forward to receiving your revised manuscript.

Kind regards,

Lakshminarayana Chekuri, MD, PhD

Academic Editor

PLOS ONE

Journal Requirements:

**Additional Editor Comments:**

Thank you for your scholarly contribution. I'd also like to thank the authors for choosing PLOS ONE to publish your findings from this study. Comments from reviewers are provided below. Please review these comments and I suggest address them and resubmit your manuscript. Your timely response would help this study be published and will make it accessible to interested readers across the world. I look forward to reviewing your revised manuscript. I wish you good luck with your future endeavors.

Reviewers' comments:

Reviewer's Responses to Questions

**Comments to the Author**

1. If the authors have adequately addressed your comments raised in a previous round of review and you feel that this manuscript is now acceptable for publication, you may indicate that here to bypass the “Comments to the Author” section, enter your conflict of interest statement in the “Confidential to Editor” section, and submit your "Accept" recommendation.

Reviewer #1: (No Response)

2. Is the manuscript technically sound, and do the data support the conclusions?

Reviewer #1: Yes

3. Has the statistical analysis been performed appropriately and rigorously? 

Reviewer #1: Yes

4. Have the authors made all data underlying the findings in their manuscript fully available?

Reviewer #1: Yes

5. Is the manuscript presented in an intelligible fashion and written in standard English?

Reviewer #1: No

6. Review Comments to the Author

Reviewer #1: Peer Review Report

Manuscript Title: Cultural Stigma, Psychological Distress and Help-Seeking: Moderating role of Self-esteem and Self-stigma in Lebanon

To the Authors:

Thank you for submitting the revised manuscript. The research topic is of significant interest and the study has the potential to make a valuable contribution to the field.

To help the manuscript reach its full potential, this review focuses on issues of language and formatting that, if addressed, will greatly enhance the clarity and professionalism of the work. It is important to note that the manuscript still contains several basic errors that can be resolved with a thorough proofread.

To ensure the scientific merit of the work is presented as clearly as possible, comprehensive proofreading by the authors is essential. This allows the peer review process to focus on the scientific content rather than on copy-editing. To that end, this review provides a detailed list of language and formatting issues that remain. We strongly encourage the authors to not only correct these specific points but also to conduct a proactive and comprehensive review of the entire manuscript.

A thoroughly proofread manuscript will allow reviewers to fully engage with the important research presented and will facilitate a smooth path toward final acceptance.

Specific Issues for Correction

A. Spelling and Typographical Errors

1.Location: In the ‘Limitations and Future Research’ subsection.

Original Sentence: “To the best ofour knowledge, this is the first study addressing stigma, self-stigma and self- esteem in a sample of young adults with existing mental disorders.”

Problem: ‘ofour’ is a typo and should be corrected to ‘of our’.

2.Location: In the ‘Focus on the Lebanese Context’ subsection.

Original Sentence: “Lebanon is a small Levantine country with a unique collectivist culture and a heterogenous landscape of 18 officially recognized religious groups (Faour, 2007).”

Problem: ‘heterogenous’ is a typo and should be corrected to ‘heterogeneous’.

B. Grammar and Sentence Structure

3.Location: In the ‘BACKGROUND’ section, under the ‘Stigma, Self-Stigma and Mental Distress’ subsection.

Original Sentence: “A significant body of research on mental health stigma and its deleterious impacts have for instance explained how the perception of individuals with schizophrenia as dangerous and commonly crazy have, among other, lead to internalized negative beliefs about oneself referred to as self-stigma (Yanos et al., 2015).”

Problem: The subject is “A significant body of research” (singular), so the verb must also be singular (has), not have. This is a subject-verb agreement error.

4.Location: In the ‘Measures’ subsection, in the description of the Self-Stigma Questionnaire (SSQ).

Original Sentence: “The scale systematically shows high level of internal consistency, and Cronbach's alpha coefficients ranging from 0.75 to 0.901”

Problem: The part of the sentence after and is a sentence fragment; it lacks a main verb and breaks the parallel structure.

5.Location: In the ‘DISCUSSION’ section.

Original Sentence: “The relationship between stigma, psychological distress and help-seeking intentions is further complicated and influenced by self-esteem as well as self- stigma, that are shown to play crucial mediating roles respectively.”

Problem: For this non-restrictive clause, ‘which’ should be used instead of ‘that’. The use of that after a comma is incorrect.

6.Location: In the ‘Conclusion’ section.

Original Sentence: “Feedback, from focus group interviews post-workshop showed that the program succeeded in instilling favorable attitudes and beliefs about themselves...”

Problem: The comma after "Feedback" is unnecessary and incorrectly separates the subject from its verb, "showed," making the sentence structure awkward.

C. Wording and Clarity

7.Location: In the ‘Procedure’ subsection.

Original Sentence: “The link was subsequently sponsored by targeting Lebanese 18-40 years users.”

Problem: The phrasing is unidiomatic. It should be revised to the more standard “users aged 18-40” or “18- to 40-year-old users.”

8.Location: In the ‘Statistical Analysis’ subsection.

Original Sentence: “...since the skewness (= .001) and kurtosis (= -.702) values varied between -1 and +1 (Field, 2013).”

Problem: The word “varied”is used imprecisely here, as skewness and kurtosis are two fixed values. Using “were between -1 and +1” or “fell within -1 and +1 ” would be more accurate.

9.Location: In the ‘RESULTS’ section, under the ‘Bivariate analysis of factors associated with attitude help seeking’ subsection.

Original Sentence: “The Bonferroni post-hoc analysis results showed that, in terms of age, there was a significance difference between the group aged 20-29 years and the following age groups...”

Problem: The adjective “significant” should be used here to modify the noun difference, not the noun “significance”.

D. Internal Consistency

10.Location: In the ‘Focus on the Lebanese Context’ subsection and the ‘References’ section.

Problem: The text refers to the “National Mental Health Strategy for Lebanon”as covering the years (2023-2030), but the title of the strategy listed in the references section indicates the years are (2024- 2030). These dates are inconsistent and must be verified and unified.

Final Recommendation

Based on the detailed review above, my final recommendation is:

Minor Revision

7. PLOS authors have the option to publish the peer review history of their article (what does this mean? ). If published, this will include your full peer review and any attached files.

**Do you want your identity to be public for this peer review?** For information about this choice, including consent withdrawal, please see our Privacy Policy .

Reviewer #1: No

---

## [Author Response · Author response to Decision Letter 5]

23 Jun 2025

Dearest Editorial team

On behalf of my colleagues, we are grateful for the comments provided to help uplift our manuscript and meet journal criteria.

We are hereby providing a point-by-point reply to the editor and reviewers in blue color; and have highlighted in yellow tracked changes in the manuscript for ease of tracking. We have also conducted a thorough proofread of the manuscript.

Thank you for giving us the opportunity to give visibility to our work from the underrepresented part of the world.

To the Authors:

Thank you for submitting the revised manuscript. The research topic is of significant interest and the study has the potential to make a valuable contribution to the field. To help the manuscript reach its full potential, this review focuses on issues of language and formatting that, if addressed, will greatly enhance the clarity and professionalism of the work. It is important to note that the manuscript still contains several basic errors that can be resolved with a thorough proofread. To ensure the scientific merit of the work is presented as clearly as possible, comprehensive proofreading by the authors is essential. This allows the peer review process to focus on the scientific content rather than on copy-editing. To that end, this review provides a detailed list of language and formatting issues that remain.

We strongly encourage the authors to not only correct these specific points but also to conduct a proactive and comprehensive review of the entire manuscript. A thoroughly proofread manuscript will allow reviewers to fully engage with the important research presented and will facilitate a smooth path toward final acceptance. Done. Thank you for your encouragement and support.

Specific Issues for Correction

A. Spelling and Typographical Errors

1.Location: In the ‘Limitations and Future Research’ subsection.

Original Sentence: “To the best ofour knowledge, this is the first study addressing stigma, self-stigma and self- esteem in a sample of young adults with existing mental disorders.”

Problem: ‘ofour’ is a typo and should be corrected to ‘of our’. Thank you. This typo is now corrected in the manuscript.

2.Location: In the ‘Focus on the Lebanese Context’ subsection.

Original Sentence: “Lebanon is a small Levantine country with a unique collectivist culture and a heterogenous landscape of 18 officially recognized religious groups (Faour, 2007).”

Problem: ‘heterogenous’ is a typo and should be corrected to ‘heterogeneous’. Thank you. This typo is now corrected in the manuscript.

B. Grammar and Sentence Structure

3.Location: In the ‘BACKGROUND’ section, under the ‘Stigma, Self-Stigma and Mental Distress subsection.

Original Sentence: “A significant body of research on mental health stigma and its deleterious impacts have for instance explained how the perception of individuals with schizophrenia as dangerous and commonly crazy have, among other, lead to internalized negative beliefs about oneself referred to as self-stigma (Yanos et al., 2015).”

Problem: The subject is “A significant body of research” (singular), so the verb must also be singular (has), not have. This is a subject-verb agreement error. Thank you. The verb “have” is now replaced by “has” in the manuscript.

4.Location: In the ‘Measures’ subsection, in the description of the Self-Stigma Questionnaire (SSQ).

Original Sentence: “The scale systematically shows high level of internal consistency, and Cronbach's alpha coefficients ranging from 0.75 to 0.901”

Problem: The part of the sentence after and is a sentence fragment; it lacks a main verb and breaks the parallel structure. “and” has been replaced by “with” in the manuscript.

5.Location: In the ‘DISCUSSION section.

Original Sentence: “The relationship between stigma, psychological distress and help-seeking intentions is further complicated and influenced by self-esteem as well as self- stigma, that are shown to play crucial mediating roles respectively.”

Problem: For this non-restrictive clause, ‘which’ should be used instead of ‘that’. The use of that after a comma is incorrect. We have replaced “that” by “which” in the manuscript.

6.Location: In the ‘Conclusion section.

Original Sentence: “Feedback, from focus group interviews post-workshop showed that the program succeeded in instilling favorable attitudes and beliefs about themselves...”

Problem: The comma after "Feedback" is unnecessary and incorrectly separates the subject from its verb, "showed," making the sentence structure awkward. The comma after “Feedback” has been removed.

C. Wording and Clarity

7.Location: In the ‘Procedure subsection.

Original Sentence: “The link was subsequently sponsored by targeting Lebanese 18-40 years users.” Problem: The phrasing is unidiomatic. It should be revised to the more standard “users aged 18-40” or “18- to 40-year-old users.” We have now replaced “18-40 years users” by “users aged 18-40”.

8.Location: In the ‘Statistical Analysis’ subsection.

Original Sentence: “...since the skewness (= .001) and kurtosis (= -.702) values varied between -1 and +1 (Field, 2013).”

Problem: The word “varied” is used imprecisely here, as skewness and kurtosis are two fixed values. Using “were” or “fell within” would be more accurate. The word “varied” has been replaced by “were”.

9.Location: In the ‘RESULTS’ section, under the ‘Bivariate analysis of factors associated with attitude help seeking subsection.

Original Sentence: “The Bonferroni post-hoc analysis results showed that, in terms of age, there was a significance difference between the group aged 20-29 years and the following age groups...”

Problem: The adjective “significant” should be used here to modify the noun difference, not the noun “significance”. We have replaced “significance” by the adjective “significant”. We apologize for this grammatical error.

D. Internal Consistency

10.Location: In the ‘Focus on the Lebanese Context’ subsection and the ‘References’ section.

Problem: The text refers to the “National Mental Health Strategy for Lebanon” as covering the years (2023-2030), but the title of the strategy listed in the references section indicates the years are (2024- 2030). These dates are inconsistent and must be verified and unified. We have verified the dates and replaced the year 2023 by 2024. We apologize for this typo error.

Final Recommendation

Based on the detailed review above, my final recommendation is: Minor Revision. Thank you

---

## [Decision Letter · Decision Letter 5]

27 Jun 2025

PONE-D-24-53040R5Cultural Stigma, Psychological Distress and Help-Seeking:

Moderating role of Self-esteem and Self-stigma in LebanonPLOS ONE

Dear Dr. El Khoury Malhame,

Thank you for submitting your manuscript to PLOS ONE. After careful consideration, we feel that it has merit but does not fully meet PLOS ONE’s publication criteria as it currently stands. Therefore, we invite you to submit a revised version of the manuscript that addresses the points raised during the review process.

We look forward to receiving your revised manuscript.

Kind regards,

Lakshminarayana Chekuri, MD, PhD

Academic Editor

PLOS ONE

Journal Requirements:

Additional Editor Comments (if provided):

Thank you for your scholarly contribution. I'd also like to thank the authors for choosing PLOS ONE to publish your findings from this study. Comments from reviewers are provided below. Please review these comments and I suggest address them and resubmit your manuscript. Your timely response would help this study be published and will make it accessible to interested readers across the world. I look forward to reviewing your revised manuscript. I wish you good luck with your future endeavors.

Reviewers' comments:

Reviewer's Responses to Questions

**Comments to the Author**

1. If the authors have adequately addressed your comments raised in a previous round of review and you feel that this manuscript is now acceptable for publication, you may indicate that here to bypass the “Comments to the Author” section, enter your conflict of interest statement in the “Confidential to Editor” section, and submit your "Accept" recommendation.

Reviewer #1: (No Response)

2. Is the manuscript technically sound, and do the data support the conclusions?

Reviewer #1: Yes

3. Has the statistical analysis been performed appropriately and rigorously? 

Reviewer #1: Yes

4. Have the authors made all data underlying the findings in their manuscript fully available?

Reviewer #1: Yes

5. Is the manuscript presented in an intelligible fashion and written in standard English?

Reviewer #1: Yes

6. Review Comments to the Author

Reviewer #1: Peer Review Report

To the Editor and Authors,

Thank you for the opportunity to review this revised manuscript. I would like to begin by commending the authors for this important and timely study. Research that gives visibility to what you've rightly called an "underrepresented part of the world" is essential for a truly global scientific discourse, and your work is a valuable contribution to this effort.

My comments below are offered in this same spirit of constructive collaboration. The study is on a strong footing, and my feedback is focused on refining the interpretation of the findings to ensure the final paper is as robust and impactful as possible.

1.On the Interpretation of Key Findings

My primary feedback centers on two key areas in the Discussion section where the interpretation appears to be disconnected from the empirical data presented in your results.

1.1The Relationship Between Age and Help-Seeking Attitudes: The discussion of age as a factor in help-seeking caught my attention. The narrative currently suggests a simpler relationship than the more nuanced picture painted by the data in Table 2. Specifically, the text in the Discussion section states: “In our sample, adults in their 30s showed more positive attitudes towards help-seeking than younger adults” and “They also showed more positive attitudes than individuals over 30”. This appears to conflict with the data, where the 40+ age group (Mean = 3.55) actually scores higher than the 30-39 group (Mean = 2.94). Revisiting this section to accurately describe and explore the complex, non-linear trend present in your data could make for an even more insightful discussion.

1.2The Moderating Role of Self-Stigma: A similar, though more subtle, issue of interpretation arises in the analysis of self-stigma's moderating role—a key finding of the study. The discrepancy seems to stem from the reverse-scoring protocol of the Self-Stigma Questionnaire (SSQ), which is correctly noted in the Methods section (“higher scores indicating lower levels of self-stigma” ). The results in Table 5 show that the negative relationship between psychological distress and help-seeking is significant at “low and moderate scores” on the SSQ. Therefore, the correct interpretation is that the effect is strongest for individuals with high and moderate levels of self-stigma. The Discussion section, however, currently draws the opposite conclusion: “…individuals experiencing high psychological distress are less likely to seek help if they also have low to moderate levels of self-stigma”. Rectifying this interpretation is crucial for the scientific accuracy of the manuscript's conclusions.

2.On Reporting Precision

On a smaller scale, a couple of points regarding reporting precision could further polish the manuscript.

2.1Alignment of the Abstract with Results: The Abstract makes a specific and compelling claim that “fear of individuals with mental illnesses, prejudice and lack of knowledge/awareness were related with internalized stigma…”. However, the Results section primarily presents analyses of the scales’ total scores and does not appear to offer a direct quantitative test of these specific components. To maintain strict scientific alignment, the conclusions in the abstract should be directly supported by the analyses presented in the body of the paper.

2.2Minor Data Consistency: A very minor numerical inconsistency was noted between the text’s report of bachelor’s degree holders (“had a bachelor degree (75.0%)” ) and the figure presented in Table 1 (74.7% ). Ensuring perfect consistency in these small details enhances the overall professional quality of the work.

3.A Note on This Detailed Review

I would also like to add a brief note about the detailed nature of this review, especially at a later stage. My intention here is wholly constructive. Peer review is a progressive process; as major issues of framing and methodology are resolved in early rounds, it becomes possible for the reviewer to focus on the finer details that elevate a good paper to an excellent one. The considerable time I have invested in this review reflects my belief in the high value and potential of your research. My goal is simply to collaborate in ensuring this important work is as robust and impactful as possible upon publication.

4.Conclusion

To reiterate, this is important and valuable research. The issues I have outlined are all addressable. I am confident that a revision that addresses these points, particularly the two core issues of data interpretation, will result in a very strong and compelling paper.

7. PLOS authors have the option to publish the peer review history of their article (what does this mean? ). If published, this will include your full peer review and any attached files.

**Do you want your identity to be public for this peer review?** For information about this choice, including consent withdrawal, please see our Privacy Policy .

Reviewer #1: No

---

## [Author Response · Author response to Decision Letter 6]

28 Jun 2025

Manuscript Title: Cultural Stigma, Psychological Distress and Help-Seeking: Moderating role of Self-esteem and Self-stigma in Lebanon

Dearest Editorial team

On behalf of my colleagues, we are grateful for the comments provided and the constructive feedback to help uplift our manuscript and meet journal criteria.

We are hereby providing a point-by-point reply to the editor and reviewers in blue color; and have highlighted in red tracked changes in the manuscript for ease of tracking.

To the Editor and Authors,

Thank you for the opportunity to review this revised manuscript. I would like to begin by commending the authors for this important and timely study. Research that gives visibility to what you've rightly called an "underrepresented part of the world" is essential for a truly global scientific discourse, and your work is a valuable contribution to this effort. My comments below are offered in this same spirit of constructive collaboration. The study is on a strong footing, and my feedback is focused on refining the interpretation of the findings to ensure the final paper is as robust and impactful as possible. We extend our sincere gratitude for your continued support and encouragement.

1.On the Interpretation of Key Findings: My primary feedback centers on two key areas in the Discussion section where the interpretation appears to be disconnected from the empirical data presented in your results.

1.1 The Relationship Between Age and Help-Seeking Attitudes: The discussion of age as a factor in help-seeking caught my attention. The narrative currently suggests a simpler relationship than the more nuanced picture painted by the data in Table 2. Specifically, the text in the Discussion section states: “In our sample, adults in their 30s showed more positive attitudes towards help-seeking than younger adults” and “They also showed more positive attitudes than individuals over 30”. This appears to conflict with the data, where the 40+ age group (Mean = 3.55) actually scores higher than the 30-39 group (Mean = 2.94). Revisiting this section to accurately describe and explore the complex, non-linear trend present in your data could make for an even more insightful discussion. We apologize for this error. We have replaced the age group “30s” by “40s” in the discussion section.

1.2The Moderating Role of Self-Stigma: A similar, though more subtle, issue of interpretation arises in the analysis of self-stigma's moderating role—a key finding of the study. The discrepancy seems to stem from the reverse-scoring protocol of the Self-Stigma Questionnaire (SSQ), which is correctly noted in the Methods section (“higher scores indicating lower levels of self-stigma” ). The results in Table 5 show that the negative relationship between psychological distress and help-seeking is significant at “low and moderate scores” on the SSQ. Therefore, the correct interpretation is that the effect is strongest for individuals with high and moderate levels of self-stigma. The Discussion section, however, currently draws the opposite conclusion: “…individuals experiencing high psychological distress are less likely to seek help if they also have low to moderate levels of self-stigma”. Rectifying this interpretation is crucial for the scientific accuracy of the manuscript's conclusions. Thank you again for your thoughtful comment. We have now corrected this mistake in the discussion section by replacing “low to moderate levels of self-stigma” by “moderate to high levels of self-stigma”.

2.On Reporting Precision: On a smaller scale, a couple of points regarding reporting precision could further polish the manuscript.

2.1 Alignment of the Abstract with Results: The Abstract makes a specific and compelling claim that “fear of individuals with mental illnesses, prejudice and lack of knowledge/awareness were related with internalized stigma…”. However, the Results section primarily presents analyses of the scales’ total scores and does not appear to offer a direct quantitative test of these specific components. To maintain strict scientific alignment, the conclusions in the abstract should be directly supported by the analyses presented in the body of the paper. We have replaced the sentence “we mostly found that fear of individuals with mental illnesses, prejudice and lack of knowledge/awareness were related with internalized stigma for those with mental health challenges” by “Stigmatization, both general and internalized, is systematically associated with decreased help-seeking. The relationship between stigma, psychological distress and help-seeking intentions is further complicated and influenced by self-esteem as well as self-stigma, which are shown to play crucial mediating roles respectively.”

2.2 Minor Data Consistency: A very minor numerical inconsistency was noted between the text’s report of bachelor’s degree holders (“had a bachelor degree (75.0%)”) and the figure presented in Table 1 (74.7% ). Ensuring perfect consistency in these small details enhances the overall professional quality of the work. We apologize again for this unintentional error. We have now replaced “75%” by “74.7%”.

3. A Note on This Detailed Review: I would also like to add a brief note about the detailed nature of this review, especially at a later stage. My intention here is wholly constructive. Peer review is a progressive process; as major issues of framing and methodology are resolved in early rounds, it becomes possible for the reviewer to focus on the finer details that elevate a good paper to an excellent one. The considerable time I have invested in this review reflects my belief in the high value and potential of your research. My goal is simply to collaborate in ensuring this important work is as robust and impactful as possible upon publication. We are deeply grateful for the reviewer's thoughtful and sound comments, and for the significant time and effort that the reviewer has invested in enhancing our manuscript.

4.Conclusion. To reiterate, this is important and valuable research. The issues I have outlined are all addressable. I am confident that a revision that addresses these points, particularly the two core issues of data interpretation, will result in a very strong and compelling paper. Thank you very much. Our team is very grateful for the time and energy investment to uplift our manuscript.

---

## [Decision Letter · Decision Letter 6]

3 Jul 2025

PONE-D-24-53040R6Cultural Stigma, Psychological Distress and Help-Seeking:

Moderating role of Self-esteem and Self-stigma in LebanonPLOS ONE

Dear Dr. El Khoury Malhame,

Thank you for submitting your manuscript to PLOS ONE. After careful consideration, we feel that it has merit but does not fully meet PLOS ONE’s publication criteria as it currently stands. Therefore, we invite you to submit a revised version of the manuscript that addresses the points raised during the review process.

We look forward to receiving your revised manuscript.

Kind regards,

Lakshminarayana Chekuri, MD, PhD

Academic Editor

PLOS ONE

Journal Requirements:

Additional Editor Comments:

Thank you for your scholarly contribution. I'd also like to thank the authors for choosing PLOS ONE to publish your findings from this study. Comments from reviewers are provided below. Please review these comments and I suggest address them and resubmit your manuscript. Your timely response would help this study be published and will make it accessible to interested readers across the world. I look forward to reviewing your revised manuscript. I wish you good luck with your future endeavors.

Reviewers' comments:

Reviewer's Responses to Questions

**Comments to the Author**

1. If the authors have adequately addressed your comments raised in a previous round of review and you feel that this manuscript is now acceptable for publication, you may indicate that here to bypass the “Comments to the Author” section, enter your conflict of interest statement in the “Confidential to Editor” section, and submit your "Accept" recommendation.

Reviewer #1: (No Response)

2. Is the manuscript technically sound, and do the data support the conclusions?

Reviewer #1: Yes

3. Has the statistical analysis been performed appropriately and rigorously? 

Reviewer #1: Yes

4. Have the authors made all data underlying the findings in their manuscript fully available?

Reviewer #1: Yes

5. Is the manuscript presented in an intelligible fashion and written in standard English?

Reviewer #1: Yes

6. Review Comments to the Author

Reviewer #1: Peer Review Report

Manuscript Title: Cultural Stigma, Psychological Distress and Help-Seeking: Mod-erating Role of Self-Esteem and Self-Stigma in Lebanon

Reviewer’s Comments:

This manuscript has undergone six rounds of revision, and I appreciate the authors’ continued efforts to address reviewer feedback. However, several key is-sues—including terminological misusage, misalignment between analyses and inter-pretations, and inconsistent claims—still remain unresolved.

These are not minor oversights but recurring concerns that impact the clarity and va-lidity of the study. To ensure real progress and avoid further delays, I strongly en-courage the authors to conduct a full, proactive review of the manuscript before resubmission. This includes carefully cross-checking all terminology, hypotheses, data interpretations, and summary statements across the entire text.

I believe the manuscript has potential, but it now requires more than point-by-point responses—it needs a self-directed, rigorous audit to reach publication standards. I look forward to reviewing a substantially strengthened version, ideally with a more detailed and reflective response letter as well.

Q1.The authors use PROCESS Macro Model 1, which is clearly a moderation model:

“The moderation analysis was conducted using PROCESS MACRO (an SPSS add-on), version 3.4, Model 1.”(Methods section)

Also:“Self-esteem, stigma and self-stigma scores were included as moderators...” (Statistical Analysis) However, the abstract and discussion incorrectly refer to “mediating roles”:“...self-esteem as well as self-stigma, which are shown to play crucial mediating roles respectively.” (Abstract and Discussion) This is a fundamen-tal mischaracterization of the statistical method, and must be corrected throughout the manuscript. Meanwhile, this sentence is not only incorrect in using “mediating,” but also misuses “respectively,” since there is no one-to-one mapping presented be-tween moderators and specific paths.

Q2. Only self-esteem is included as a moderator in the hypotheses:“H2: Self-esteem is hypothesized to moderate the relationship between psychological distress and help-seeking attitudes...” (End of Introduction) Yet self-stigma is also included in the moderation models later, without any prior theoretical framing or hypothesis. This creates a disconnect between the theoretical model and statistical analysis.

The authors are encouraged to incorporate a clear theoretical rationale for the mod-erating role of self-stigma earlier in the manuscript, ideally in the introduction, sup-ported by relevant literature. Furthermore, all hypotheses would benefit from revi-sion for greater academic clarity, specificity, and alignment with the final model.

If helpful, the authors may refer to hypothesis formulations in published studies on similar topics in PLOS ONE or related journals, particularly those addressing mod-erated relationships between psychological distress, self-stigma, self-esteem, and help-seeking (e.g., structured hypotheses that distinguish between direct, indirect, and moderated effects). This could enhance both conceptual clarity and methodo-logical consistency.

Q3. Throughout the manuscript, the terms “stigma” and “self-stigma” are at times used interchangeably:“Stigmatization, both general and internalized, is systemati-cally associated with decreased help-seeking.” (Abstract and Discussion) However, these refer to different constructs, measured with distinct scales (KSS vs. SSQ). The manuscript should consistently differentiate external cultural stigma and inter-nalized self-stigma to avoid conceptual confusion. The authors are encouraged to provide clear operational definitions of these two constructs in the introduction.

Q4. Table 2 shows that PhD (3.40) and high school (3.12) participants had higher help-seeking scores than bachelor (2.15) and master degree holders (2.37). Yet the discussion incorrectly states: “University graduates hold considerably fewer nega-tive attitudes toward therapy than both high school students and doctorate holders.” (Discussion) The authors are encouraged to revise the discussion to accurately reflect the findings.

Q5. The discussion claims: “Adults in their forties (40s) showed more positive atti-tudes towards help-seeking than younger adults.” (Discussion) However, Table 1 shows only 2 participants in this age group: “40: 2 (.8%)” (Table 1) Drawing any conclusion from such a tiny subsample is highly unreliable and should be clearly marked as non-generalizable. The authors might consider qualifying the claim more cautiously.

Q6. While the sample includes individuals from different education levels, the distribution is heavily skewed (with 74.7% being bachelor’s degree holders). Therefore, repeated references to “various educational backgrounds” (e.g. “...from various educational backgrounds and located in different Lebanese regions.” “...including those from various educational backgrounds.”“...represented various educational backgrounds.”)may risk misleading readers regarding the sample’s rep-resentativeness and the generalizability of findings. The authors are encouraged to consider softening or more precisely qualifying such statements to better reflect the actual distribution of the sample. For example, expressions such as “predominantly bachelor-level backgrounds” or “mainly undergraduate-level participants with lim-ited representation from other levels” would more accurately reflect the sample composition. It is also advisable to include this limitation in the relevant section.

Q7. Currently, the “Conclusion” section focuses solely on practice-based suggestions (e.g., engaging religious leaders, promoting drama therapy, leveraging social media), while entirely omitting a summary of empirical findings. Given that the manuscript already includes a separate Discussion/Limitations and Future Research section, this “Conclusion” does not function as such.

The authors are encouraged to restructure the final section with the following format:

New Section:

Implications and Conclusions (as a Level-1 heading), containing:

Practical Implications (shortened from current content, max 2–3 paragraphs);

Theoretical Implications (a new paragraph summarizing how findings contribute to stigma theory, help-seeking behavior models, and collectivist cultural contexts);

Conclusions (a new paragraph summarizing the study’s main findings and contributions in approximately 150–200 words).

This structure will enhance coherence, align with academic conventions, and better communicate both empirical and applied value.

7. PLOS authors have the option to publish the peer review history of their article (what does this mean? ). If published, this will include your full peer review and any attached files.

**Do you want your identity to be public for this peer review?** For information about this choice, including consent withdrawal, please see our Privacy Policy .

Reviewer #1: No

---

## [Author Response · Author response to Decision Letter 7]

29 Jul 2025

Dearest editorial team

We have hereby provided point-by point comment in bold in blue and have highlighted the tracked changes in the manuscript accordingly.

Peer Review Report

Manuscript Title: Cultural Stigma, Psychological Distress and Help-Seeking: Moderating Role of Self-Esteem and Self-Stigma in Lebanon

Reviewer’s Comments:

This manuscript has undergone six rounds of revision, and I appreciate the authors’ continued efforts to address reviewer feedback. However, several key issues—including terminological misusage, misalignment between analyses and interpretations, and inconsistent claims—still remain unresolved.

These are not minor oversights but recurring concerns that impact the clarity and validity of the study. To ensure real progress and avoid further delays, I strongly encourage the authors to conduct a full, proactive review of the manuscript before resubmission. This includes carefully cross-checking all terminology, hypotheses, data interpretations, and summary statements across the entire text. I believe the manuscript has potential, but it now requires more than point-by-point responses—it needs a self-directed, rigorous audit to reach publication standards. I look forward to reviewing a substantially strengthened version, ideally with a more detailed and reflective response letter as well.

We deeply appreciate your thorough and continued engagement with our work, by now, reviewers are practically co-authors in spirit.

While diligently working to address each point raised across the previous rounds, a bird’s-eye, integrative review was indeed warranted to smooth sentences and findings.

We have conducted a full-scale internal audit of the manuscript as per the suggestion and have:

• focused on terminological precision,

• aligned analyses and interpretations,

• capitalized on overall conceptual clarity.

Our revised submission reflects this broader self-directed effort to strengthen coherence and academic rigor. Thank you for your patience, persistence, and for nudging us from “good intentions” to “publication-ready.”

Q1. The authors use PROCESS Macro Model 1, which is clearly a moderation model:

“The moderation analysis was conducted using PROCESS MACRO (an SPSS add-on), version 3.4, Model 1.”(Methods section) Also: “Self-esteem, stigma and self-stigma scores were included as moderators...” (Statistical Analysis)

However, the abstract and discussion incorrectly refer to “mediating roles”:“...self-esteem as well as self-stigma, which are shown to play crucial mediating roles respectively.” (Abstract and Discussion) This is a fundamental mischaracterization of the statistical method and must be corrected throughout the manuscript. Meanwhile, this sentence is not only incorrect in using “mediating,” but also misuses “respectively,” since there is no one-to-one mapping presented between moderators and specific paths.

We have corrected this typo mistake and sentence fragment and have revised the abstract and manuscript for clarity. Our study indeed focuses on moderating roles and not mediation investigation. The factors of self-esteem and self-stigma have been investigated separately and not “respectively”. Thank you for the comment.

Q2. Only self-esteem is included as a moderator in the hypotheses: “H2: Self-esteem is hypothesized to moderate the relationship between psychological distress and help-seeking attitudes...” (End of Introduction) Yet self-stigma is also included in the moderation models later, without any prior theoretical framing or hypothesis. This creates a disconnect between the theoretical model and statistical analysis.

The authors are encouraged to incorporate a clear theoretical rationale for the moderating role of self-stigma earlier in the manuscript, ideally in the introduction, supported by relevant literature. Furthermore, all hypotheses would benefit from revision for greater academic clarity, specificity, and alignment with the final model.

If helpful, the authors may refer to hypothesis formulations in published studies on similar topics in PLOS ONE or related journals, particularly those addressing moderated relationships between psychological distress, self-stigma, self-esteem, and help-seeking (e.g., structured hypotheses that distinguish between direct, indirect, and moderated effects). This could enhance both conceptual clarity and methodological consistency.

The hypotheses have been edited to align with the moderation terminology used in the PROCESS analysis framework.

H1: Self-esteem will moderate the relationship between psychological distress and help-seeking attitudes, such that the negative association between psychological distress and help-seeking will be weaker at higher levels of self-esteem.

H2: Self-stigma will moderate the relationship between psychological distress and help-seeking attitudes, such that individuals with higher self-stigma may show less favorable help-seeking attitudes when experiencing psychological distress.

Q3. Throughout the manuscript, the terms “stigma” and “self-stigma” are at times used interchangeably: “Stigmatization, both general and internalized, is systematically associated with decreased help-seeking.” (Abstract and Discussion) However, these refer to different constructs, measured with distinct scales (KSS vs. SSQ). The manuscript should consistently differentiate external cultural stigma and internalized self-stigma to avoid conceptual confusion. The authors are encouraged to provide clear operational definitions of these two constructs in the introduction.

The intent of the study is to indeed introduce the 2 different concepts of stigma and self-stigma. The operational definition has been highlighted in the introduction in the segment on stigma, self-stigma, and mental distress, and further revisited throughout the entire manuscript.

Yet, at times, the authors point out that both work in the same direction, as was set in the abstract in relation for example with diminished help-seeking attitudes. This has been changed to “Stigmatization, in both forms i.e., external/cultural, and internalized,” to avoid confusion.

Q4. Table 2 shows that PhD (3.40) and high school (3.12) participants had higher help-seeking scores than bachelor (2.15) and master degree holders (2.37). Yet the discussion incorrectly states: “University graduates hold considerably fewer negative attitudes toward therapy than both high school students and doctorate holders.” (Discussion) The authors are encouraged to revise the discussion to accurately reflect the findings.

Thank you for pointing out this. After revising the discussion section, we have removed this paragraph to avoid confusing the reader as the bivariate analysis is not the main focus of this study.

Q5. The discussion claims: “Adults in their forties (40s) showed more positive attitudes towards help-seeking than younger adults.” (Discussion) However, Table 1 shows only 2 participants in this age group: “40: 2 (.8%)” (Table 1) Drawing any conclusion from such a tiny subsample is highly unreliable and should be clearly marked as non-generalizable. The authors might consider qualifying the claim more cautiously.

We removed this paragraph since it discusses results from the bivariate analysis, which is not the main focus of this study.

Q6. While the sample includes individuals from different education levels, the distribution is heavily skewed (with 74.7% being bachelor’s degree holders). Therefore, repeated references to “various educational backgrounds” (e.g. “...from various educational backgrounds and located in different Lebanese regions.” “...including those from various educational backgrounds.”“...represented various educational backgrounds.”)may risk misleading readers regarding the sample’s representativeness and the generalizability of findings. The authors are encouraged to consider softening or more precisely qualifying such statements to better reflect the actual distribution of the sample. For example, expressions such as “predominantly bachelor-level backgrounds” or “mainly undergraduate-level participants with limited representation from other levels” would more accurately reflect the sample composition. It is also advisable to include this limitation in the relevant section.

Thank you for your thoughtful comment. We have replaced “various educational backgrounds” in the abstract and Participants’ section by “predominantly bachelor-level backgrounds” and added the following sentence in the limitation section.

Our study indeed focused on educated young adults and should thus be generalized with care. Future studies should consider involving individuals with varying levels of education and broader age ranges.

Q7. Currently, the “Conclusion” section focuses solely on practice-based suggestions (e.g., engaging religious leaders, promoting drama therapy, leveraging social media), while entirely omitting a summary of empirical findings. Given that the manuscript already includes a separate Discussion/Limitations and Future Research section, this “Conclusion” does not function as such.

The authors are encouraged to restructure the final section with the following format:

New Section:

Implications and Conclusions (as a Level-1 heading), containing:

Practical Implications (shortened from current content, max 2–3 paragraphs);

Theoretical Implications (a new paragraph summarizing how findings contribute to stigma theory, help-seeking behavior models, and collectivist cultural contexts);

Conclusions (a new paragraph summarizing the study’s main findings and contributions in approximately 150–200 words).

This structure will enhance coherence, align with academic conventions, and better communicate both empirical and applied value.

This is a very helpful recommendation. The authors have actioned it accordingly. The practical implications reflect steps within the community whereas the theoretical implications call to upgrade the mentioned stigma theories. Lastly the conclusion sums up ways forward in local, regional, and global context.

Practical Implications

This study sheds light on the significance of mental health stigma on psychological distress and help-seeking. Findings highlight the pivotal role of both personal and social targets in addressing the matter, especially given that self-esteem and self-stigma were found to moderate the above intricate relationships. In a collectivist Lebanese context, this means capitalizing on social connectedness by intensifying early awareness within appropriate cultural norms.

At a local level, contact with community leaders, known as Mukhtars, and religious figures to address stigma, is essential to bridge the gap between the Non-Governmental Organizations (NGOs) mental health workers and communities. Those can be seen as outsiders who come to change the traditional norms, especially in a sectarian country such as Lebanon (Osman & Chebaro, 2024). To this extent, culturally sensitive interventions that promote mental health literacy, and challenge misconceptions should be advocated along local leaders including religious clergy; whom have the trust of the community and have a crucial role in referring those who might initially seek help from religious figures or attribute mental health issues to spiritual factors (Osman & Chebaro, 2024).

Other actionable steps would include actively fostering youngsters’ self-esteem within families, societies, schools, universities, and municipalities with specific evidence-based programs aimed at improving self-esteem and reducing stigma. One such program significantly enhanced emotional intelligence skills in public high-school students, while also targeting self-esteem and self-stigma (Sanchez-Ruiz et al., 2023). Focus group interviews conducted after the workshop showed that the program effectively instilled favorable self-attitudes and beliefs about addressing psychological distress. In particular, the program helped participants manage and overcome feelings of inferiority and shame concerning mental disorders and individuals experiencing them.

Another practical measure involves using social media platforms such as WhatsApp and Instagram to facilitate open dialogue about mental health and build public empathy. This approach has been supported by studies in stigmatized collectivists nations such as Lebanon (Osman & Chebaro, 2024) and Malaysia (Rahman et al., 2022). To this extent, people who might have depression or experience anxiety can connect to other Lebanese people, and to mental health specialists and in turn override feelings of loneliness, isolation, and helplessness.

Theoretical Implications

The present study provides important theoretical insights into the interplay between stigma, help-seeking behavior, and psychological distress within a collectivist cultural context. Our findings substantiate and extend Stigma Theory by providing evidence that both public and internalized stigma are not only prevalent but deeply interwoven with self-perceptions and psychological suffering in such contexts. The moderating roles of self-esteem and self-stigma confirm key tenets of Stigma Management Theory, particularly the idea that personal resources and internalized beliefs shape individuals’ coping strategies in the face of societal rejection.

Finally, in collectivist cultures like Lebanon, where family honor and communal belonging heavily influence identity, our findings suggest that inasmuch as social values offer community support (Osman & Chebaro, 2024), they can negatively influence help-seeking as stigma carries a heightened threat in some social circles. To avoid worsening such internal conflicts, this study urges culturally adapted theories of mental health behavior that account for communal interdependence and honor-based identity systems. Promising success of interventions building on comedy and group cohesion such as drama therapy have also shown efficacy and cultural sensitivity in coming together to heal and overcome stigma (Haddad et al., 2024). These would be particularly relevant as they would factor in cultural resources such as the reported collective sense of humor, used to laugh at shared miseries and to cope with the country’s latest socio- economic and political tensions.

Conclusion

This study contributes to the limited but growing literature on mental health stigma in the Arab world by examining how stigma, psychological distress, and individual-level moderators such as self-esteem and self-stigma shape help-seeking attitudes in a Lebanese sample. Findings could be relevant in the Global south faced with protracted crises and in the Global North with incrementally rising rates of mental distress. We build on recommendations from the Lancet Commission on ending stigma and discrimination in mental health (2022), suggesting nuanced internal processes may either buffer or compound the effects of stigma. Since approaching people with direct lived experiences of psychological disorders, our research could inform policy makers in comparable Levantine cultures to increase help-seeking attitudes by capitalizing on positive mental health literacy to build self-esteem and contribute to individual-resilience, while incentivizing proactive efforts to collectively destigmatize and treat mental illness. This would lead to favorable outcomes for both mental and physical health of vulnerable young adults.

Best regards,

Corresponding author

---

## [Decision Letter · Decision Letter 7]

12 Aug 2025

PONE-D-24-53040R7Cultural Stigma, Psychological Distress and Help-Seeking: Moderating role of Self-esteem and Self-stigma in LebanonPLOS ONE

Dear Dr. El Khoury Malhame,

Thank you for submitting your manuscript to PLOS ONE. After careful consideration, we feel that it has merit but does not fully meet PLOS ONE’s publication criteria as it currently stands. Therefore, we invite you to submit a revised version of the manuscript that addresses the points raised during the review process.

We look forward to receiving your revised manuscript.

Kind regards,

Lakshminarayana Chekuri, MD, PhD

Academic Editor

PLOS ONE

Journal Requirements:

Additional Editor Comments ;

Thank you for your scholarly contribution. I'd also like to thank the authors for choosing PLOS ONE to publish your findings from this study. This manuscript is nearing acceptance pending some minor corrections suggested by the reviewer. Comments from reviewer are provided below. Please review these comments and I suggest address them and resubmit your manuscript. Your timely response would help this study be published and will make it accessible to interested readers across the world. I look forward to reviewing your revised manuscript. I wish you good luck with your future endeavors.

Reviewers' comments:

Reviewer's Responses to Questions

**Comments to the Author**

1. If the authors have adequately addressed your comments raised in a previous round of review and you feel that this manuscript is now acceptable for publication, you may indicate that here to bypass the “Comments to the Author” section, enter your conflict of interest statement in the “Confidential to Editor” section, and submit your "Accept" recommendation.

Reviewer #1: All comments have been addressed

2. Is the manuscript technically sound, and do the data support the conclusions?

Reviewer #1: Yes

3. Has the statistical analysis been performed appropriately and rigorously? 

Reviewer #1: Yes

4. Have the authors made all data underlying the findings in their manuscript fully available?

Reviewer #1: Yes

5. Is the manuscript presented in an intelligible fashion and written in standard English?

Reviewer #1: Yes

6. Review Comments to the Author

Reviewer #1: Thank you for your patience and effort. After seven rounds of detailed revisions, I am pleased to see that the manuscript's overall quality, logical rigor, and clarity of argument have all improved very significantly. Based on these important improvements, I will be recommending acceptance to the editorial office.

Before the manuscript proceeds to the next stage, I would like to suggest two final, minor points for polishing to ensure the work is presented as professionally as possible:

1. In the discussion on the moderating role of self-stigma, a sentence reads: "Within this framework, we hypothesize that self-stigma may serve as a protective factor...". This statement is quite confusing as it appears to directly contradict your findings. According to your data in Table 5, high levels of self-stigma actually function as a risk factor, exacerbating the negative relationship between psychological distress and help-seeking attitudes. I would recommend thoroughly revising this sentence to ensure it aligns with your data.

2. The manuscript uses a numerical citation system (e.g., [68]), but in some places, this is combined with an author-year format, such as in "...when subjected to stigma [68] (El Hayek et al., 2021).". This is redundant. I suggest a final check of the manuscript to remove such redundancies, leaving only the numerical citations for consistency and adherence to the journal's style.

7. PLOS authors have the option to publish the peer review history of their article (what does this mean? ). If published, this will include your full peer review and any attached files.

**Do you want your identity to be public for this peer review?** For information about this choice, including consent withdrawal, please see our Privacy Policy .

Reviewer #1: No

---

## [Author Response · Author response to Decision Letter 8]

13 Aug 2025

Reviewer #1: Thank you for your patience and effort. After seven rounds of detailed revisions, I am pleased to see that the manuscript's overall quality, logical rigor, and clarity of argument have all improved very significantly. Based on these important improvements, I will be recommending acceptance to the editorial office.

We are even more thankful for the patience and the grit in making this publication happen. A Million thanks

Before the manuscript proceeds to the next stage, I would like to suggest two final, minor points for polishing to ensure the work is presented as professionally as possible:

1. In the discussion on the moderating role of self-stigma, a sentence reads: "Within this framework, we hypothesize that self-stigma may serve as a protective factor...". This statement is quite confusing as it appears to directly contradict your findings. According to your data in Table 5, high levels of self-stigma actually function as a risk factor, exacerbating the negative relationship between psychological distress and help-seeking attitudes. I would recommend thoroughly revising this sentence to ensure it aligns with your data.

We rephrased the idea as suggested. Thank you.

2. The manuscript uses a numerical citation system (e.g., [68]), but in some places, this is combined with an author-year format, such as in "...when subjected to stigma [68] (El Hayek et al., 2021).". This is redundant. I suggest a final check of the manuscript to remove such redundancies, leaving only the numerical citations for consistency and adherence to the journal's style.

Corrected.

We would like to thank all the editorial team again and again for all your efforts and time.

This manuscript has allowed us to grow in ways we never imagined <3

---

## [Editor Report · Decision Letter 8]

26 Aug 2025

Cultural Stigma, Psychological Distress and Help-Seeking:

Moderating role of Self-esteem and Self-stigma in Lebanon

PONE-D-24-53040R8

Dear Dr. El Khoury Malhame,

We’re pleased to inform you that your manuscript has been judged scientifically suitable for publication and will be formally accepted for publication once it meets all outstanding technical requirements.

Kind regards,

Lakshminarayana Chekuri, MD, PhD

Academic Editor

PLOS ONE
---

## [Editor Report · Acceptance letter]

PONE-D-24-53040R8

PLOS ONE

Dear Dr. El Khoury Malhame,

I'm pleased to inform you that your manuscript has been deemed suitable for publication in PLOS ONE. Congratulations! Your manuscript is now being handed over to our production team.

Kind regards,

on behalf of

Dr. Lakshminarayana Chekuri

Academic Editor

PLOS ONE